# Modulating FOXO3 transcriptional activity by small, DBD-binding molecules

Judith Hagenbuchner[1†], Veronika Obsilova[2†], Teresa Kaserer[3,4,5†], Nora Kaiser[4], Bettina Rass[4], Katarina Psenakova[2,6], Vojtech Docekal[7], Miroslava Alblova[2], Klara Kohoutova[2,6], Daniela Schuster[3,8], Tatsiana Aneichyk[9,10], Jan Vesely[7], Petra Obexer[1,5], Tomas Obsil[2,6*], Michael J Ausserlechner[4*]

[1]Department of Pediatrics II, Medical University Innsbruck, Innsbruck, Austria; [2]Department of Structural Biology of Signaling Proteins, Institute of Physiology, Division BIOCEV, The Czech Academy of Sciences, Prague, Czech Republic; [3]Pharmaceutical Chemistry, Institute of Pharmacy, University of Innsbruck, Innsbruck, Austria; [4]Department of Pediatrics I, Medical University Innsbruck, Innsbruck, Austria; [5]Tyrolean Cancer Research Institute, Innsbruck, Austria; [6]Department of Physical and Macromolecular Chemistry, Faculty of Science, Charles University, Prague, Czech Republic; [7]Department of Organic Chemistry, Faculty of Science, Charles University, Prague, Czech Republic; [8]Department of Pharmaceutical and Medicinal Chemistry, Paracelsus Medical University Salzburg, Salzburg, Austria; [9]Division of Molecular Pathophysiology, Biocenter, Medical University Innsbruck, Innsbruck, Austria; [10]Independent Data Lab UG, Munich, Germany

*For correspondence:
obsil@natur.cuni.cz (TO);
michael.j.ausserlechner@i-med.ac.
at (MJA)

†These authors contributed
equally to this work

Competing interests: The
authors declare that no
competing interests exist.

Reviewing editor: Xavier
Darzacq, University of California,
Berkeley, United States

**Abstract** FOXO transcription factors are critical regulators of cell homeostasis and steer cell death, differentiation and longevity in mammalian cells. By combined pharmacophore-modeling-based in silico and fluorescence polarization-based screening we identified small molecules that physically interact with the DNA-binding domain (DBD) of FOXO3 and modulate the FOXO3 transcriptional program in human cells. The mode of interaction between compounds and the FOXO3-DBD was assessed *via* NMR spectroscopy and docking studies. We demonstrate that compounds S9 and its oxalate salt S9OX interfere with FOXO3 target promoter binding, gene transcription and modulate the physiologic program activated by FOXO3 in cancer cells. These small molecules prove the druggability of the FOXO-DBD and provide a structural basis for modulating these important homeostasis regulators in normal and malignant cells.

## Introduction

The mammalian forkhead box O (FOXO) transcription factor family (FOXO1/FKHR, FOXO3/FKHRL1, FOXO4/AFX, and FOXO6) is involved in multiple cellular processes ranging from apoptosis induction to longevity (*Hornsveld et al., 2018*). In mammals, FOXO3 and its family members recognize and bind the same core DNA elements (TTGTTTAC) to control the transcription of direct target genes. However, despite sharing the same consensus sequences, FOXO members serve distinct functions and may act as tissue-specific homeostasis regulators.

FOXO3 was initially considered as a tumor suppressor that induces apoptosis and cell-cycle arrest (*Calnan and Brunet, 2008*). However, its functions in cellular detoxification (*de Keizer et al., 2011; Salcher et al., 2014*), and drug-resistance (*Hui et al., 2008; Wilson et al., 2011*), maintenance of cancer stem cells (*Naka et al., 2010*) as well as inhibition of other death-inducers such as TP53 (*Rupp et al., 2017*) suggest also a tumor-promoting role in certain types of cancer.

Multiple cellular signaling pathways converge on FOXO3, most importantly the pro-proliferative PI3K-PKB pathway that inactivates target gene transcription. On the other hand, stress conditions, such as genotoxic stress, reactive oxygen species (ROS) or hypoxic stress, override growth-factor-mediated phosphorylation of FOXO3, which results in the relocation of FOXO3 to the nucleus (*Hagenbuchner et al., 2012*; *Hagenbuchner et al., 2013*). Thereby, stress induced signaling kinases, such as JNK or MST1 that cause FOXO3 activation and nuclear accumulation also in presence of PKB signaling critically contribute to FOXO3-triggered therapy-resistance programs that protect cancer cells during therapy (*Hagenbuchner et al., 2016b*; *Rupp et al., 2017*; *Naka et al., 2010*). In addition, FOXO3 regulates the differentiation of naïve regulatory T-cells *via* its transcriptional target FOXP3 (*Kerdiles et al., 2010*; *Harada et al., 2010*), which limits the cytotoxic anti-cancer T-cell response by immune-suppressive regulatory T-cells that infiltrate tumor tissue. A reversible inhibition of FOXO3 activity by small compounds thereby might boost anti-tumor immune responses and limit side effects of FOXO3 functional inactivation.

In contrast to the small, defined substrate binding pockets on catalytic enzymes, the DNA binding domain (DBD) of transcription factors are usually regarded as 'undruggable' due to the large surfaces and the fact that the only known ligand is a DNA molecule. Small molecules have been described for the transcription factor FOXM1 (*Gormally et al., 2014*; *Hegde et al., 2011*) and one compound was shown to regulate FOXO1 activity (*Nagashima et al., 2010*), but no compounds have been discovered that directly physically interact with the DBD of FOXO proteins to regulate their transcriptional activity. By a pharmacophore model-based, virtual in silico screening approach we identified compound S9 and demonstrate that this molecule inhibits FOXO3-binding to target promoters, affects the cell-wide transcriptional program of FOXO3, as well as FOXO3 effects on cellular ROS-production and cell growth in 2D and 3D cell culture systems. By NMR we demonstrate that S9 directly interacts with FOXO3 DBD, elucidate the mode of binding and how this molecule structurally interferes with FOXO3 transcriptional activity.

## Results

### Identification of compound S9 as FOXO-DBD ligand

In the absence of known small molecule ligands, a structure-based modeling workflow (*Figure 1a*) was developed, employing the crystal structure of FOXO3 DBD in complex with a 13 bp FOXO3 consensus sequence DNA strand (PDB entry 2UZK; *Tsai et al., 2007*). The site to be targeted within the large interaction surface was defined using experimentally observed FOXO3-DNA interactions, consensus sites predicted by four pocket prediction algorithms (*Kulharia et al., 2009*; *Le Guilloux et al., 2009*; Pocket-Finder Pocket Detection (http://www.modelling.leeds.ac.uk/pocketfinder/), Molecular Operating Environment (MOE, https://www.chemcomp.com/), literature data on crucial residues and less flexible sites as suggested by mutational studies and posttranslational modifications (*Tsai et al., 2007*) and a molecular dynamics simulation on the related FOXO4 (*Boura et al., 2007*), respectively. Due to the limited amount of available data, we aimed to elucidate interaction patterns of potential ligands by combining data from the interactions observed in the protein-DNA complex (*Figure 1b*), interaction hotspots on the protein surface calculated with MOE (*Figure 1c*), and binding modes predicted by docking of Drugbank (*Wishart et al., 2008*) version 2.5 into the binding site (*Figure 1d*). The identified binding patterns were represented by six pharmacophore models, which were subsequently used for virtual screening of the Specs (www.specs.net) and May-bridge (www.maybridge.com) databases. 76 virtual hits for which the desired binding mode was confirmed by further docking studies were selected for experimental testing. Compound S9 (1-(4,6-dimethylpyrimidin-2-yl)−3-(4-propoxyphenyl)guanidine) was identified by pharmacophore model 1 (*Figure 1e*).

Primary biochemical validation of hit-compounds was performed via fluorescence polarization analyses (FPA) using recombinant FOXO3-DBD (residues 156–269) and FAM-labeled oligonucleotides containing the IRE consensus sequence (CTA TCA AAA CAA CGC) (*Figure 1—figure supplement 1a*) in combination with flow cytometric live/dead analyses (*Figure 1—figure supplement 1bc*). Assay variation of FPA was assessed by calculating Z′ factor (*Figure 1—figure supplement 2d*: Z′factor = 0.6499). As shown in *Figure 1f*, compounds S9 reduced the interaction of FOXO3-DBD to the FAM-IRE oligonucleotide in a dose-dependent manner. We further synthesized an oxalate salt

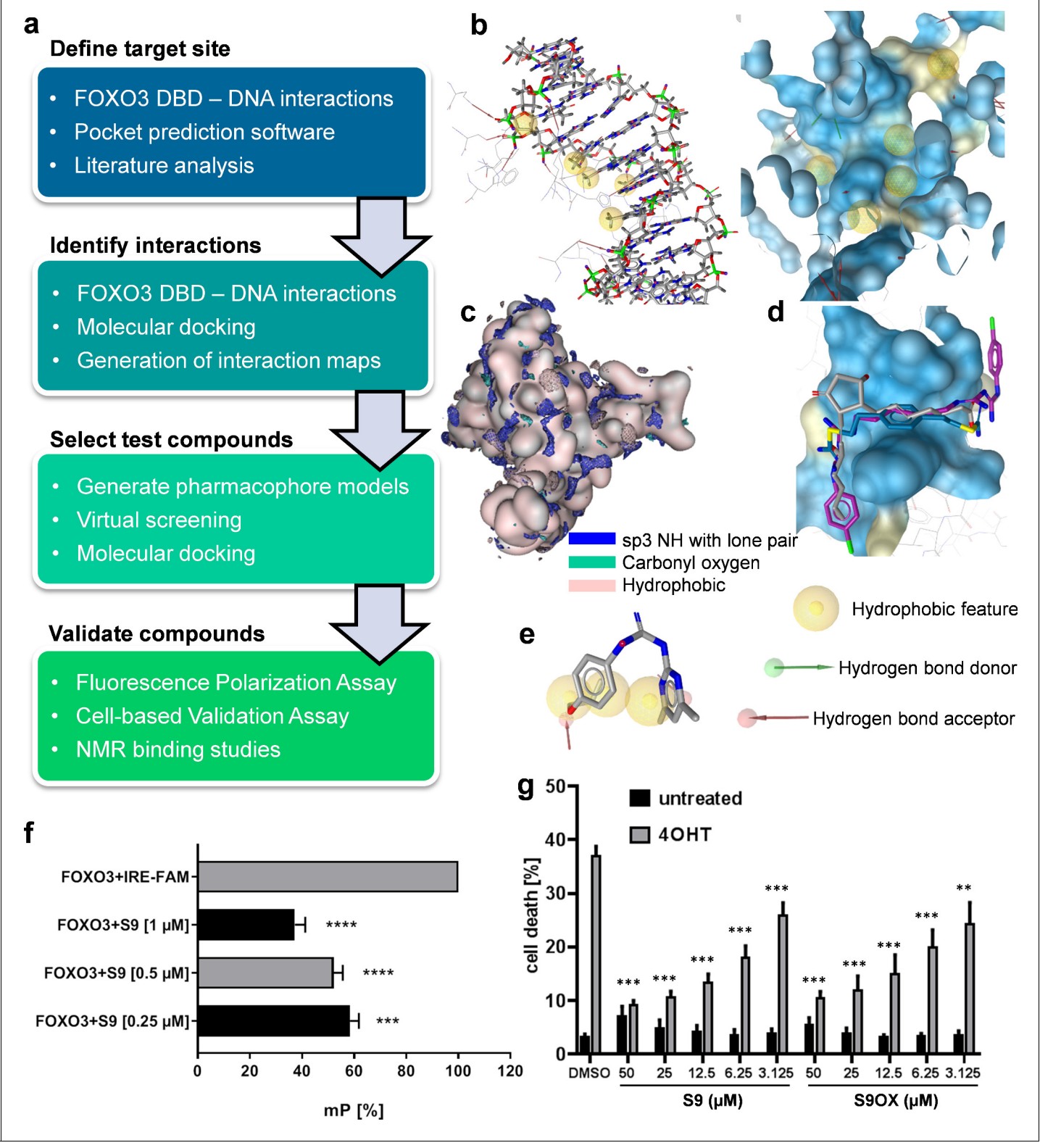

**Figure 1.** Strategy to identify small molecule compounds that interact with FOXO3-DBD. (**a**) Overview of the workflow employed to identify FOXO3 inhibitors. (**b**) Interactions between FOXO3-DBD and the 13 bp DNA strand were represented as pharmacophore features. DNA is shown as sticks and FOXO3 DBD residues as lines (left) and the pharmacophore features were mapped onto the FOXO3 DBD protein surface (blue, right). (**c**) Interaction maps highlight areas on the FOXO3 DBD surface (depicted in rose) where the defined probes can interact with the protein. (**d**) Docking poses of

*Figure 1 continued on next page*

*Figure 1 continued*

DB00878 (magenta), DB02056 (gray), and DB02141 (blue) were used to generate different pharmacophore models. (**e**) Compound S9 maps to one of the pharmacophore models. (**f**) FPA of recombinant FOXO3-DBD (125 nM) and FAM-labeled IRE-oligonucleotide (25 nM). S9 dose-dependent increase of freely rotating oligonucleotide is demonstrated. Shown is the mean of three independent experiments + SD (****p<0.0001, ***p<0.001). (**g**) PI-staining of nuclei and flow cytometric analyses of SH-EP/FOXO3 cells treated with 50 nM 4OHT alone or in combination with varying concentrations of S9 or S9OX for 48 hr. Shown is the mean + SD of four independent experiments. Statistical differences between 4OHT and S9+4OHT or 4OHT and S9OX+4OHT were assessed by students t-test (***p<0.001, **p<0.01, two-tailed).

The online version of this article includes the following figure supplement(s) for figure 1:

**Figure supplement 1.** Overview of combined fluorescence polarization assay (FPA)- and live/dead flow cytometry-based validation of candidate compounds.

**Figure supplement 2.** Effect of S9 on protein-DNA interaction of different FOX family members.

(S9OX) of S9 to improve the solubility of this compound in water (*Supplementary file 1*). To validate these biochemical screening results in living cells we used SH-EP/FOXO3 neuroblastoma cells stably expressing a 4-hydroxy-tamoxifen (4OHT)-regulated FOXO3(A3)ERtm transgene that undergo apoptotic cell death upon activation of FOXO3 (*Obexer et al., 2007*; *Hagenbuchner et al., 2016b*). As shown in *Figure 1g* (and *Figure 1—figure supplement 1bc*), the addition of compound S9 and S9OX interfered with FOXO3-induced cell death in a dose-dependent manner to similar extent, demonstrating that S9 and the oxalate salt S9OX enter living cells and modulate FOXO3 function. Based on these biochemical and cell-based results we defined S9 as a promising FOXO3-modulatory compound worth for further biochemical, cell biological and structural characterization.

## Mapping of the S9-binding site in FOXO3-DBD

Structural characterization of the interaction between FOXO3-DBD and S9 was performed using NMR spectroscopy. First, $^1$H saturation transfer difference (STD) NMR measurements were used to assess S9 binding to FOXO3-DBD. STD signals were detected for several protons of both S9 and its oxalate salt, thus confirming their interaction with FOXO3-DBD (*Figure 2a* and *Figure 2—figure supplement 1a*). In the case of S9OX, pronounced STD signals were observed for H1, H3, H5, and H7 protons, whereas less pronounced signals were observed for H4 and H6 protons. In the case of S9, whose STD spectra were acquired in the presence of 10% DMSO due to its low solubility in water, strong STD signal was observed for H3 protons and less pronounced signals were recorded for H1, H5, H6 and H7 protons. These data suggested that both aromatic moieties as well as the aliphatic part of S9 are involved in direct interactions with FOXO3-DBD.

Because the previously published NMR sequential assignment of human FOXO3-DBD (*Wang et al., 2008*) was obtained for shorter construct (residues 151–251) compared to the construct used in this study (residues 156–269), standard triple resonance experiments were used to obtain a sequence specific backbone assignment of our FOXO3-DBD$_{156-269}$ construct. The $^1$H-$^{15}$N heteronuclear single quantum coherence (HSQC) spectrum of $^{15}$N-labeled FOXO3-DBD with obtained resonance assignment is shown in *Figure 2—figure supplement 1b*. The data analysis provided a resonance assignment for 99 out of 114 residues (87% of the FOXO3-DBD sequence). To identify the S9-binding site in FOXO3-DBD, the $^{15}$N-labeled FOXO3-DBD was titrated with S9OX and $^1$H and $^{15}$N chemical shift perturbations (CSPs) of the backbone amide groups of FOXO3-DBD were followed in $^1$H-$^{15}$N HSQC spectra (*Figure 2—figure supplement 1c*). The oxalate salt of S9 was used for its higher solubility in water, thus avoiding the use of DMSO and the associated chemical interferences. The presence of S9OX induced significant dose-dependent CSPs of backbone amide groups of fifteen FOXO3-DBD residues (the chemical shift change was greater than $\sigma^0_{corr}$ above the mean), thus suggesting their involvement in the interaction or their conformational change induced by S9 binding (*Figure 2b*). When these residues were mapped onto the solution structure of FOXO3-DBD (*Wang et al., 2008*), they revealed the binding surface for S9 in the region formed by the DNA recognition α-helix H3 and the N-terminal part of β-strand S2 (*Figure 2c*). The most affected residues (the chemical shift change was greater than $2\sigma^0_{corr}$ above the mean) were Arg211, His212, Asn213, Ser215 and Leu216 from the helix H3; Phe220 and Asn237 from the wing W2; and Thr167 from the helix H1. In addition, the gradual shift of resonances during titration indicated fast exchange of the ligand on the NMR time scale, thus suggesting that the S9OX binding is of moderate affinity.

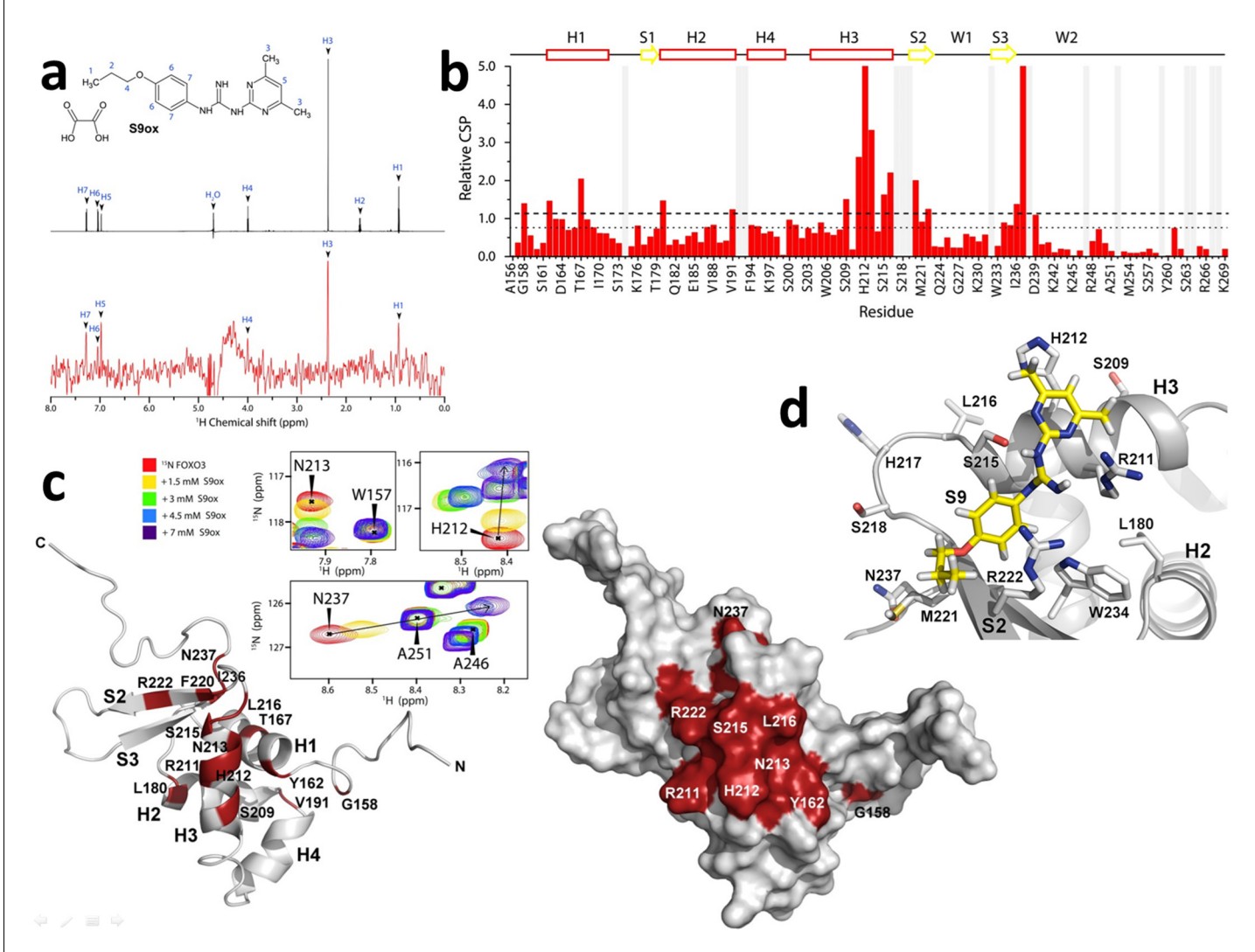

**Figure 2.** Compounds S9/S9OX block the DNA binding surface of FOXO3-DBD. (a) 1D $^1$H STD-NMR experiments for S9OX compound in the presence of the 15 μM FOXO3-DBD. The reference spectrum of S9OX is shown in black, the corresponding STD-NMR spectrum is shown in red. Hydrogens in S9 are numbered from 1 to 7. (b) The summary of quantified chemical shift perturbations (CSPs) obtained for the FOXO3-DBD in the presence of 1.5 mM S9OX. The changes in chemical shift resonances were calculated using weighted combination of chemical shifts given by: $CSP = \sqrt{\Delta\delta_H^2 + \left(\frac{1}{5}\Delta\delta_N\right)^2}$, where $\Delta\delta_H$ and $\Delta\delta_N$ are differences in chemical shifts of $^1$H and $^{15}$N, respectively, in the free and bound states (*Williamson, 2013*). The relative CSPs were obtained by dividing absolute CSP values by a standard deviation of the whole set of CSPs. The regions of the protein backbone that could not be unambiguously assigned are highlighted in gray. The secondary structure of FOXO3-DBD is indicated on top. The dotted and dashed lines indicate changes greater than the mean and the mean + $1\sigma^0_{corr}$, respectively. (c) Observed CSPs mapped onto the solution structure of FOXO3-DBD (*Wang et al., 2008*). Ribbon and surface representations are shown at right and left, respectively. Thirteen residues whose $^1$H-$^{15}$N resonances were significantly perturbed ($1\sigma^0_{corr}$ above the mean) are shown in red. Selected signals from the $^1$H-$^{15}$N HSQC spectra of 250 μM FOXO3-DBD in the presence of various concentrations of S9OX are shown in inset. (d) Structural model of FOXO3-DBD with bound S9. The best flexible docking solution consistent with data from $^1$H-$^{15}$N HSQC and STD measurements is shown.

The online version of this article includes the following figure supplement(s) for figure 2:

**Figure supplement 1.** S9 binding to FOXO3-DBD.
**Figure supplement 2.** Comparison of S9OX binding to various FOXO proteins.

To obtain greater insight into the interaction of S9 with FOXO3-DBD, docking calculations using Autodock Vina (*Trott and Olson, 2010*) were performed. The S9 binding surface (the residues whose side-chains were treated as flexible) included residues identified in $^1$H-$^{15}$N HSQC experiment as well as several additional residues with high CSPs located close to them (*Figure 2d*). The

calculated poses with the lowest binding energy were extracted and further validated against results from STD NMR measurements. The final pose is consistent with results from both STD and $^1$H-$^{15}$N HSQC experiments and suggests that S9 binds into the groove formed by residues Arg211, His212 and Ser215 at the C-terminus of α-helix H3 and residues Met221 and Arg222 from the N-terminus of β-strand S2. This region is a crucial part of the FOXO3-DBD/DNA interface as the DNA recognition α-helix H3 is responsible for most of the direct base contacts with DNA and the N-terminal part of β-strand S2 is involved in contacts with the DNA backbone (*Tsai et al., 2007*). Therefore, these data indicate that S9 binding blocks the DNA binding surface of FOXO3-DBD in the region of the recognition α-helix H3.

## Compounds S9 and S9OX affect the FOXO3 transcriptional program

To analyze the effect of compound S9 on mRNA steady state levels of FOXO3-regulated genes we assessed genome-wide regulatory events by mRNA expression profiling. SH-EP/FOXO3 cells were treated with 50 nM 4OHT for three hours in presence or absence of 50 μM compound S9 and the transcriptome was measured with Affymetrix U133 2.0 whole genome chips. 1262 probesets were induced and 889 probesets showed statistically significant repression upon activation of ectopic FOXO3(A3)ERtm as demonstrated in heatmaps (*Figure 3a*) and volcano blots (*Figure 3b*). When 4OHT-driven activation of ectopic FOXO3 was combined with S9, the gene induction was much less pronounced, with the expression of only 208 probesets upregulated of which 35 were common with FOXO3 activation alone (*Figure 3c*). We examined whether the presence of S9 reversed FOXO3-mediated gene induction and identified 193 probesets showing significant repression in presence of S9, suggesting an inhibition of FOXO3-mediated induction (*Figure 3b*). Furthermore, 779 probesets have shown the reversal of expression although they did not reach statistical significance. This suggests that S9 inhibited FOXO3-mediated induction of the majority of mRNAs (77% of FOXO3 induced probesets overall, 15% statistically significant). We also observed an increase in steady state levels of 173 probesets that were not regulated by ectopic FOXO3. These changes in mRNA expression might result from differences in target gene regulation between ectopic FOXO3(A3)ERtm and endogenous FOXO3, as ectopic FOXO3(A3)ERtm carries mutations at the major PKB phosphorylation sites (T32, S253 and S315) or off-target effects. This effect of S9 is well visualized in the heatmap analysis (*Figure 3a*) where S9 attenuates the strong induction of a large cluster of genes (top-part of heatmaps) or represses highly expressed FOXO3 targets by its own (bottom part of heat map), suggesting inhibition of endogenous FOXO3. Interestingly, S9 alone also induced a cluster of strongly repressed FOXO3-targets (middle part of heatmaps) and, for a small number of genes, S9 increased FOXO3-mediated induction (arrows). When comparing S9 with DMSO control, 412 probesets were repressed. Therefore, although S9 mainly exerts an inhibitory effect on FOXO3 transcriptional activity, this compound also induces mRNA expression of a small number of FOXO3 target genes.

To verify genome-wide gene expression results we selected four genes previously identified as FOXO3 targets in neuroblastoma cells and evaluated the effect of compound S9 and its oxalate salt S9OX on mRNA expression by quantitative RT-PCR analyses. As shown in *Figure 3d–g*, activation of ectopically-expressed FOXO3(A3)ERtm by 4OHT significantly induced steady state expression of the pro-apoptotic BH3-only proteins BIM and NOXA, as well as of the detoxifying proteins SESN3 and DEPP1 similar to previously published data from our group (*Hagenbuchner et al., 2012*; *Obexer et al., 2007*; *Salcher et al., 2014*; *Salcher et al., 2017*). S9 or S9OX alone caused no statistically significant changes in steady state expression of these genes compared to solvent-treated controls. However, when combined with activation of ectopic FOXO3, S9 as well S9OX significantly counteracted mRNA induction of all four *bona fide* FOXO3 targets suggesting that this compound efficiently interferes with specific mRNA induction by FOXO3.

## FOXO3-modulatory compounds S9 and S9OX inhibit the induction of FOXO3 target proteins, binding of FOXO3 to target promoters and promoter transactivation

Next, we quantified the effects of compound S9 and S9OX by immunoblot analyses to assess whether regulations observed on mRNA steady state level were also translated into changes in protein expression. As shown in *Figure 4a–d* the activation of the 4OHT-regulated FOXO3(A3)ERtm transgene in SH-EP/FOXO3 neuroblastoma cells elevated the protein levels of the proteins BIM,

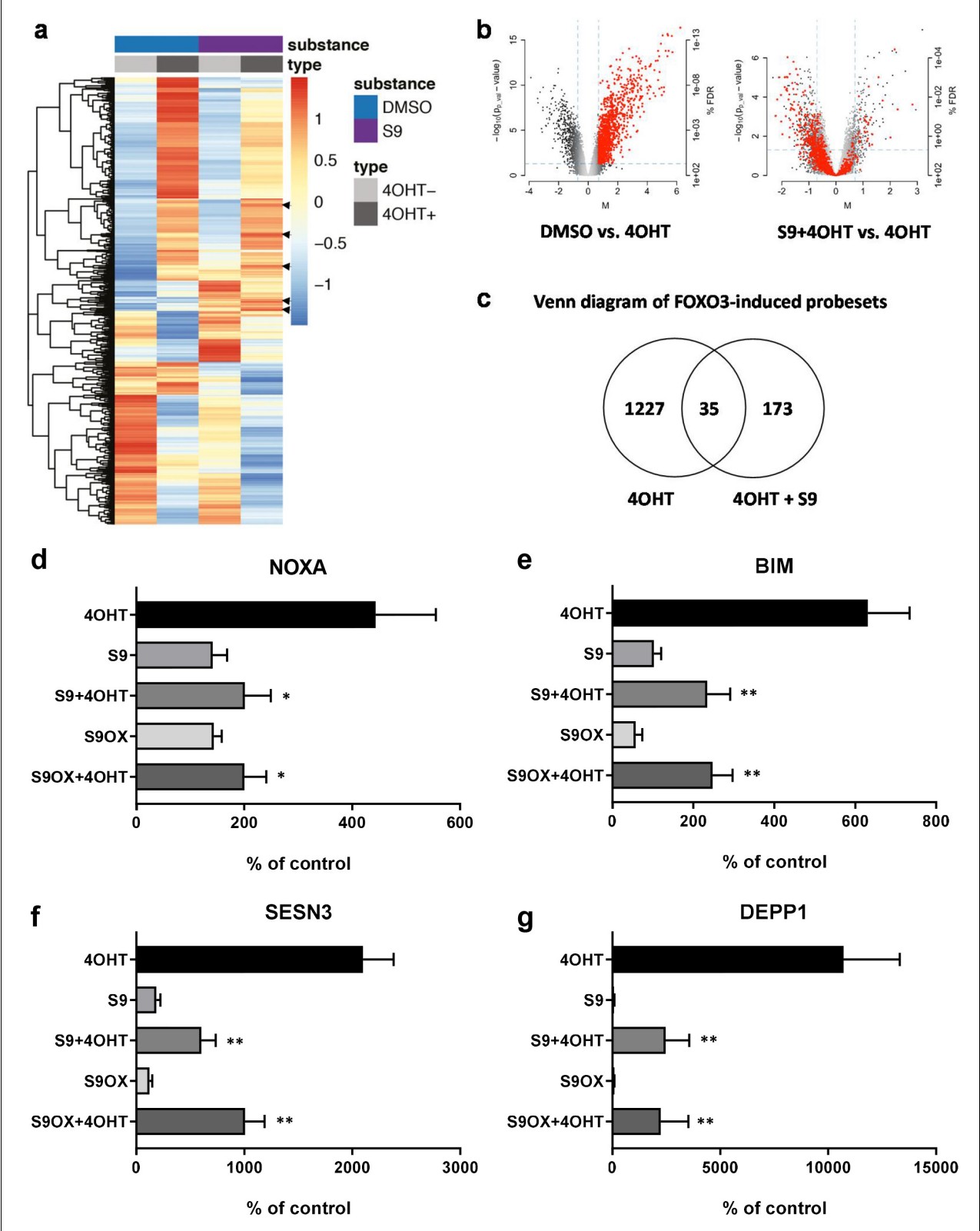

**Figure 3.** Compounds S9/S9OX affect induction of FOXO3 target gene mRNAs. (a) Heatmaps of Affymetrix microarray analyses (U133 plus 2.0 expression profiling chips). Total RNA was prepared of SH-EP/FOXO3 cells treated with 100 nM 4OHT alone or in combination with 50 μM S9 for three hours. Expression heatmap shows average expression of the differentially expressed genes within three replicates in each condition. The rows were scaled around 0. (b) Left volcano blot shows log2 fold changes of all probesets on the microarray on x-axis and the significance level (-log10(p-value))
*Figure 3 continued on next page*

*Figure 3 continued*

on the y-axis – probesets significantly induced by FOXO3 are in red. The right volcano blot shows log2 fold change of S9+4OHT as compared to FOXO3-induced probesets. (c) Venn diagram showing numbers of genes regulated in common when cells with activated FOXO3 were treated with S9 or solvent as control. (d) NOXA, (e) BIM, (f) SESN3, and (g) DEPP1 mRNA levels were measured by quantitative RT-PCR in SH-EP/FOXO3 cells after treatment with 100 nM 4OHT for three hours alone or in combination with 50 µM S9 or S9OX (cells were preincubated with compounds for 30 min). Bars represent mean + SD of three independent experiments, each performed in triplicates (untreated control was set as 100%). Significant differences between 4OHT treatment and S9+4OHT or S9OX+4OHT: *p<0.05, **p<0.01 (students t-test, two-tailed).

NOXA, SESN3 and DEPP1. Consistent with quantitative RT-PCR results in *Figure 3d–g*, both S9 and S9OX prevented FOXO3-mediated induction of NOXA, BIM and SESN3 above background level and significantly attenuated the strong induction of DEPP1, demonstrating that S9 and S9OX efficiently interfere with the regulation of FOXO3 target proteins in living cells. The combined data suggest that S9 and S9OX both modulate FOXO3-transcriptional program and target protein regulation. Biochemical results from FPA and NMR-based structural analyses suggest that in vitro the compounds S9/S9OX bind to the DBD of FOXO3 and inhibit DBD – DNA interaction. To address, whether S9 directly interferes with FOXO3 target promoter recognition in living cells we next performed chromatin immunoprecipitation (ChIP) analyses and assessed the binding of FOXO3 to the promoters of BIM, NOXA, SESN3 and DEPP1. As demonstrated in *Figure 4e*, activation of the conditional FOXO3 allele for three hours significantly increased FOXO3 promoter binding, which is consistent with the previously demonstrated induction of these FOXO3 targets on mRNA and protein level. Importantly, for all investigated promoters, the addition of S9 almost completely reduced the interaction of FOXO3 with endogenous promoters to control level. To assess, whether this reduced promoter binding also correlates with reduced transactivation we used a luciferase reporter vector containing a 544 bp DEPP1 promoter fragment that carries three putative FOXO3 consensus sites (*Salcher et al., 2014*). As demonstrated in *Figure 4f*, compound S9 reduced luciferase activity in a dose-dependent manner with approximately 50% reduction at 12.5 µM. These results confirm the biochemical binding studies between compounds S9 and S9OX and FOXO3-DBD (*Figure 1f, Figure 1—figure supplement 1a* and *Figure 2*) and demonstrate that S9 inhibits binding of FOXO3 to target promoters and their activation in living cells.

## Selectivity of compound S9/S9OX for FOXO transcription factors

In above experiments we demonstrated that the in silico identified S9 compound physically interacts with FOXO3-DBD and affects binding to FOXO3 responsive promoters as well as FOXO3 gene regulation in a cell system with a conditionally activated ectopic FOXO3 allele. We next investigated whether S9 can still modulate FOXO3 effects in a cell system with a 4OHT-activated FOXO3(A3)-H212R-ERtm mutant. H212 is strongly affected by binding of S9 (*Figure 2b*) suggesting that this amino acid critically participates in S9 binding. As demonstrated in *Figure 5a* activation of FOXO3(A3)-H212R still induced apoptotic cell death in SH-EP cells although to a significantly lower extent than the FOXO3(A3) construct. However, in contrast to FOXO3(A3)-induced cell death, S9 did not affect apoptosis by the FOXO3(A3)-H212R mutant suggesting that the H212 mutation completely abrogates FOXO3-inhibitory effects of S9. As to this point all experiments were done using ectopically expressed, conditional FOXO3(A3) alleles we next tested whether S9 might also interfere with the regulation of target genes by endogenous FOXO3 in response to chemotherapeutics and/or serum withdrawal. Using CRISPR/Cas9 technology we generated FOXO3 knockout cell lines that completely lack endogenous FOXO3, transiently transfected luciferase reporter plasmids for BIM and DEPP1 and treated these cells with etoposide (after serum withdrawal) to activate endogenous FOXO3 (*Hagenbuchner et al., 2012*). As shown in *Figure 5bc*, etoposide treatment increases promoter activity of the BIM and the DEPP1 promoter two- to threefold and S9 completely prevents this induction. Knockout of endogenous FOXO3 abrogates the effect of etoposide treatment on FOXO3-responsive promoters and S9 does not further affect promoter activity suggesting that the inhibitory effect of S9 depends on the presence of endogenous FOXO3 protein and is not due to unspecific effects on transcription. To assess this question further on endogenous expression of FOXO3 target genes we analyzed expression levels of BIM, DEPP1 and SESN3 by qRT-PCR under conditions of etoposide treatment and/or serum withdrawal (0.5% FBS). Serum withdrawal significantly increased expression of DEPP1 and SESN3 and additional treatment with etoposide strongly

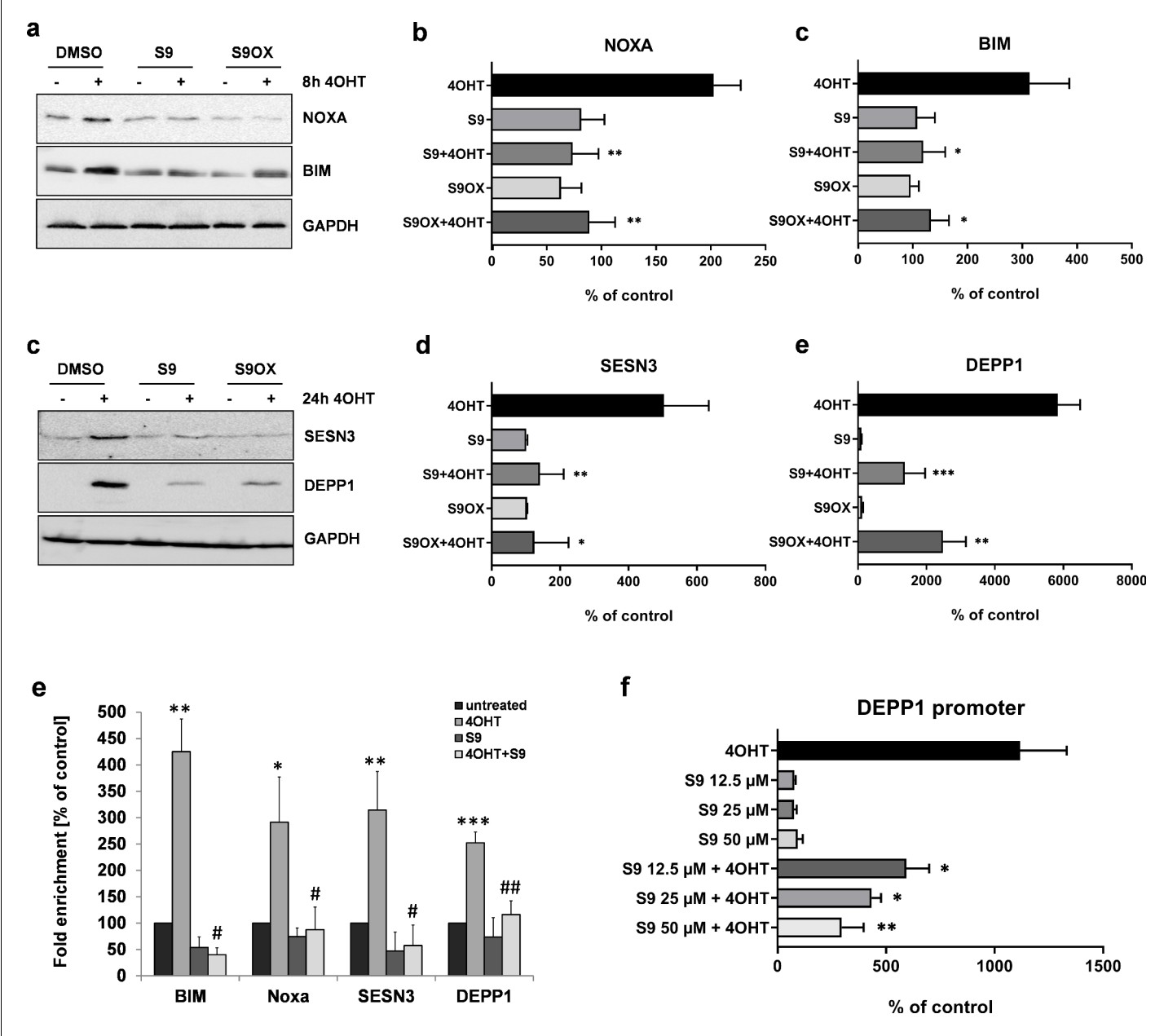

**Figure 4.** Compounds S9 and S9OX inhibit protein expression of FOXO3-regulated proteins and prevent binding of FOXO3 to target promoters. SH-EP/FOXO3 cells were treated for eight hours (a,b) or 24 hr (c,d) with 100 nM 4OHT alone or in combination with 50 μM S9 or S9OX. Cell lysates were subjected to immunoblot analyses using antibodies against NOXA and BIM (8 hr) or SESN3 and DEPP1 (24 hr). Shown are representative immunoblots (a,c) or densitometric analyses of three independent cell lysates mean values + SD. Regulations are expressed as fold over DMSO-control (100%). Significant differences between 4OHT treatment and S9+4OHT or S9OX+4OHT were analyzed by students t-test: *p<0.05, **p<0.01, ***p<0.001; two-tailed. (e) ChIP analyses were performed in SH-EP/FOXO3 cells treated with 100 nM 4OHT for three hours alone or in combination with 50 μM S9. Binding of FOXO3 to the promoter regions of BIM, NOXA, SESN3, and DEPP1 was measured by quantitative PCR. Shown is the mean value + SD of three independent experiments, each performed in duplicates. Significantly different to untreated cells: *p<0.05, **p<0.01, ***p<0.001, two-tailed. (f) Binding of FOXO3 to the promoter region of DEPP1 was assessed after transfection of a DEPP1-luciferase reporter plasmid into SH-EP/FOXO3 cells. 24 hr after transfection cells were seeded into 24 well plates. After adherence for another 24 hr, cells were treated for three hours with 100 nM 4OHT with or without increasing amounts of S9 (preincubated for 30 min). Firefly-luciferase was analyzed using the Luciferase Assay System (Promega). The increase of light emission (relative light units, RLU) was calculated as percent of untreated controls. Shown are mean values + SD of four independent experiments, each performed in triplicate. Significant differences between 4OHT treatment and S9+4OHT: *p<0.05, **p<0.01 (students t-test, two-tailed).

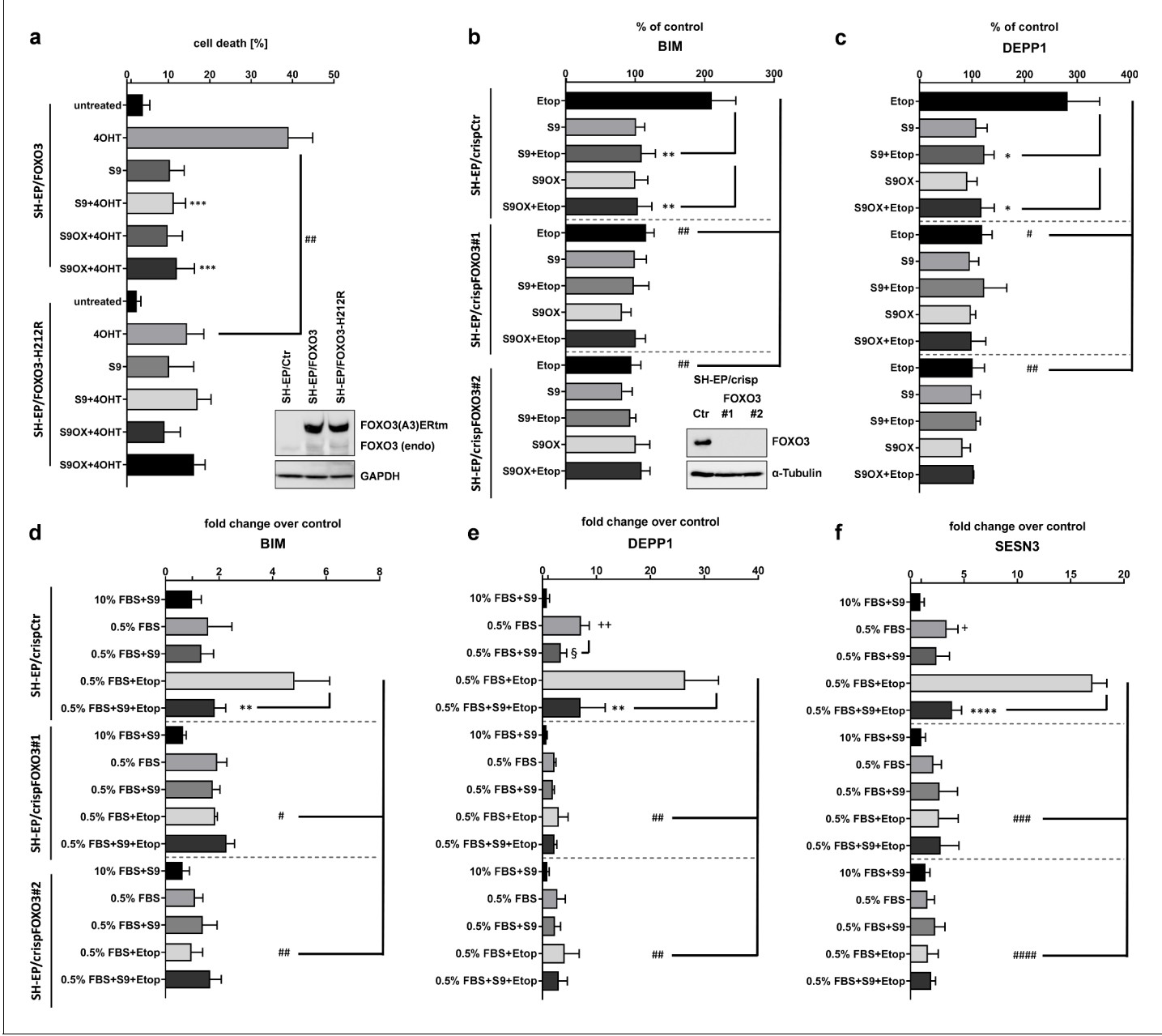

**Figure 5.** Selectivity assessment of S9 and S9OX for FOXO3. SH-EP cells expressing a conditionally activated FOXO3(A3)-ERtm-H212R mutant or FOXO (A3)-ERtm were analyzed by immunoblot, treated with 20 nM 4OHT with or without 50 µM S9 or S9OX and subjected to flow cytometric cell death analyses after 48 hr (a). Shown is the mean of four independent experiments (two-tailed students t-test: ##p<0.01, ***p<0.001). To assess whether S9- or S9OX-mediated inhibition of FOXO target gene regulation relies on endogenous FOXO3 we generated FOXO3 knock-out cells using CRISPR/Cas9 technology. Immunoblot demonstrates complete loss of endogenous FOXO3 expression in lines #1 and #2. SH-EP/crispCtr and SH-EP/crispFOXO3#1 and #2 cells were transfected with either a Bim-promoter-, a DEPP1-promoter-, or control luciferase reporter plasmid. 24 hr after transfection cells were seeded into 24 wells and kept under 0.5% FBS for 16 hr. Pre-incubation (30 min) with 50 µM S9 or S9OX was performed before 20 µg/ml etoposide were applied for three hours. Luciferase-activity of Bim- and DEPP1-promoters was assessed using Luciferase Assay System (Promega) according to manufacturer's instructions. RLU were normalized to control luciferase reporter and expressed as % of control. Shown is the mean of three independent experiments (two-tailed students t-test: #p<0.05, ##p<0.01, *p<0.05, **p<0.01; b, c). SH-EP/crispCtr and SH-EP/crispFOXO3#1 and #2 cell lines were kept at 10% FCS or 0.5% FCS for 16 hr, treated for 30 min with 50 µM of compounds S9 and then for another three hours with 20 µg/ml etoposide as indicated. Effect of endogenous expression of Bim (d), DEPP1 (e) and SESN3 (f) was assessed by quantitative RT-PCR. Shown is the mean of three independent experiments each performed in triplicate (two-tailed students t-test: #p<0.05, ##p<0.01, ###p<0.001, +p<0.05, ++p<0.01, *p<0.05, **p<0.01, §<0.05).

induced the expression of all three target genes. In FOXO3 knockout cells, still some increase of DEPP1 and SESN3 expression was observed which was not affected by S9. The strong induction of all three genes by a combination of serum-withdrawal and etoposide treatment, which was efficiently inhibited by S9 was also almost completely abrogated by FOXO3 knockout. The combined data clearly demonstrate that S9 transcriptional inhibition of FOXO target genes depends on the presence of cellular FOXO3. To further assess the selectivity of S9 within forkhead proteins and in specific FOXO family members in vitro we established and performed FP assays with recombinant FOXO1-DBD (159-272), FOXO3-DBD, FOXO4-DBD (82-207) and FOXO6-DBD (87-200) as well as with FOXM1-DBD (222 - 360). As demonstrated in *Figure 1—figure supplement 2a*, at a concentration of 250 nM, S9 most efficiently displaced the FAM-labelled IRE-oligonucleotide from FOXO3-DBD and FOXO4-DBD and had a less pronounced effect on FOXO1-DBD. Interestingly, the substance did not interfere with FOXO6-DBD – DNA binding at all. Differential binding/selectivity for FOXO-family members might be explained by the fact that although the amino acid sequences of FOXO-DBDs are highly homologous, NMR solution structures reveal significant differences between the DBDs of these closely related proteins (*Psenakova et al., 2019*). To assess whether S9 might exert effects by binding to other important forkhead proteins such as FOXM1 we also established an FP-assay for the FOXM1-DBD, analyzed S9 effects at 500 nM and observed no effect on FOXM1-DBD – DNA interaction (*Figure 1—figure supplement 2b*). To further complement these analyses with direct binding studies we performed NMR spectroscopy measurements which demonstrated differential effects of S9OX on $^1$H and $^{15}$N chemical shift perturbations of backbone amide groups in $^{15}$N-labeled FOXO1-DBD, FOXO3-DBD and FOXO4-DBD proteins (*Figure 2—figure supplement 2*).

## Compounds S9 and S9OX prevent ROS-induction and cell death by FOXO3 in 2D and 3D cell culture models

We previously demonstrated that FOXO3-activation induces a bi-phasic reactive oxygen species (ROS) wave that critically contributes to induction of apoptotic cell death in human neuroblastoma cells (*Hagenbuchner et al., 2012*). To assess the effect of S9 and S9OX on ROS accumulation by FOXO3 we either activated ectopic FOXO3(A3)ERtm by 4OHT treatment (*Figure 6a*) or treated the cells with 20 μg/ml etoposide to activate endogenous FOXO3 *via* DNA-damage response (*Figure 6b*). Direct activation of ectopic FOXO3 by 4OHT or etoposide treatment cause significant accumulation of cellular ROS as demonstrated by live-cell fluorescence microscopy using the ROS-sensitive dye reduced mitotracker red. Both S9 and S9OX completely prevent ROS accumulation in 4OHT- and etoposide-treated cells demonstrating that S9 and S9OX efficiently prevent cellular ROS accumulation in response to FOXO3 activation. Since results obtained with 2D cell culture only partially reflect effects of compound treatment in tissues or solid tumors we assessed the effect of S9 and its oxalate salt S9OX in 3D spheroids of neuroblastoma cells formed by magnetic bioprinting and magnetic levitation culture. For these experiments the FOXO3-sensitive, high-stage neuroblastoma cell line NB15/FOXO3-GFP was used to directly image cell growth/viability by fluorescence microscopy. To define at which concentrations S9 and S9OX are still biologically active and how long physiologic effects may be observed upon single treatment we reduced the concentrations of S9 and S9OX to 5 μM. Tumor spheres grown for 72 hr floating in a magnetic field were treated with 10 nM 4OHT for 72 hr in presence or absence of S9 and S9OX and then cultured for another week. After this time, sphere-size and number of spheres per well were analyzed by fluorescence microscopy (*Figure 6c–e*). Metabolic activity and cell viability were assessed by measuring ATP-content (*Figure 6f*) and reduction of resazurin salt (*Figure 6g*), respectively. Both, S9 and S9OX prevented FOXO3-induced cell death in 3D tumor spheroids at a concentration of 5 μM and after single administration. Although sphere diameter was slightly reduced in S9/S9OX + 4OHT-treated cells compared to controls, still normal numbers of well-formed tumor spheres were present. Importantly, ATP-content and resazurin salt reduction, which are both parameters for cell viability, demonstrated that S9 and S9OX preserved the viability of tumor spheres and strongly inhibited the apoptosis-inducing effect of FOXO3 in this 3D tumor model. The combined data demonstrate that S9 and its oxalate salt S9OX physically interact with the FOXO3-DBD, inhibit FOXO3 transcriptional program and modulate FOXO3 physiologic function in the low micromolar range in living cells.

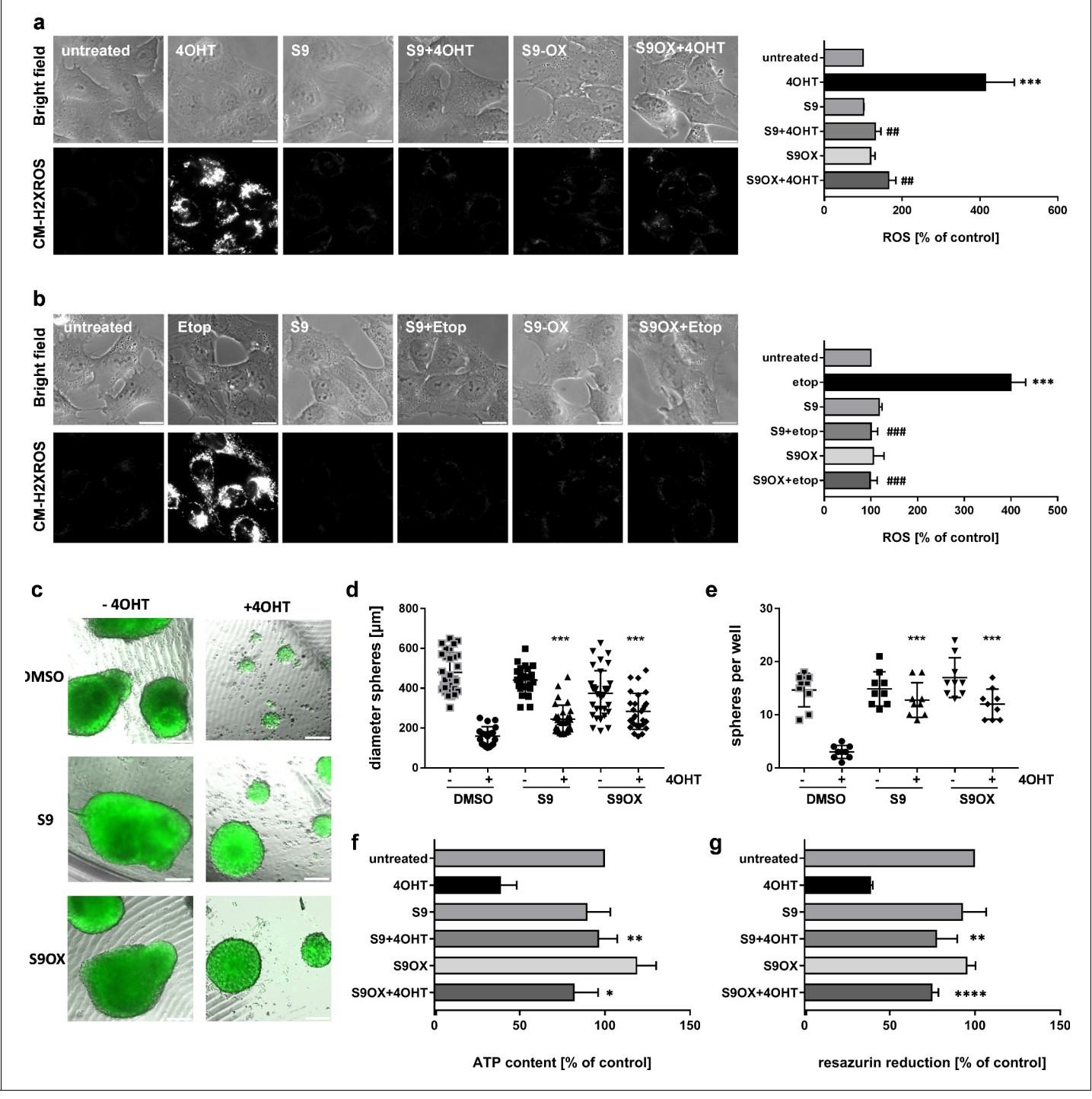

**Figure 6.** Effects of compounds S9/S9OX on FOXO3-induced ROS and sphere growth in 3D cell culture models. SH-EP/FOXO3 cells were treated either for four hours with 100 nM 4OHT (**a**) or for two hours with 20 µg/ml etoposide (**b**). 50 µM S9 or S9OX were pre-incubated for 15 min. ROS accumulation was analyzed using CM-H2XROS (500 nM). Images were acquired by live-cell imaging using an Axiovert200M microscope, equipped with a 63x oil objective, bar 20 µm. Shown are representative images (left panel) or densitometric analyses of three independent experiments (more than 15 cells per experiment were quantified) using AxioVision software version 4.8; Significantly different to untreated cells (students t-test, two-tailed): ***p<0.001; significantly different between 4OHT-treatment and S9+4OHT or S9OX+4OHT cells: ##p<0.01, ###p<0.001; significantly different between etop-treatment and S9+etop or S9OX+etop cells: ###p<0.001. 3D spheroids of NB15/FOXO3-GFP cells that express a 4OHT-regulated FOXO3 allele and constitutively EGFP were formed by magnetic bioprinting for 72 hr before treatment with 10 nM 4OHT alone or in combination with 5 µM S9 or S9OX for another 72 hr. After one week, spheroids were analyzed by live cell microscopy and further used for viability analyses. Shown are representative images (**c**) and sphere size/mean diameter (**d**) as well as the number of spheroids per well (**e**) out of three independent experiments.

*Figure 6 continued on next page*

*Figure 6 continued*

Statistical differences between single 4OHT treatment or combinational treatment were assessed by Mann-Whitney test (two-tailed ***$P<0.001$). Viability of spheroids was assessed by measuring of ATP content (f) and resazurin reduction (g). Shown are mean values + SD of three independent experiments each performed with spheres of eight different wells. Statistical differences were assessed by students t-test (two-tailed) between 4OHT-DMSO and 4OHT-compound (*$P<0.05$, **$P<0.01$, ****$p<0.0001$).

## Discussion

In this manuscript we describe a small compound that inhibits FOXO3 transcriptional activity by direct binding to the FOXO3-DBD. Based on structural data of FOXO3 and FOXO4 we developed six different pharmacophore-models that were then used for in silico screening of structural compound databases. Candidate compounds were tested biochemically by FPA and for biological efficacy in a cell system stably expressing a conditional FOXO3(A3)ERtm allele that can be activated by the addition of 4OHT. We identified compounds that hamper FOXO3-DNA interaction and enter living cells. We selected S9 which also shows increased binding to FOXO3 within FOXO family members (*Figure 1—figure supplement 2* and *Figure 2—figure supplement 2*) for further investigation due to its relatively high solubility in water, established the whole chemical synthesis pathway and synthesized an oxalate salt of S9 with increased solubility in water (*Supplementary file 1*).

NMR measurements together with molecular docking simulations suggest that S9 binds to the pocket formed by the C-terminal part of α-helix H3 and the wing W1 (*Figure 2c and d*). This region is a crucial part of the DNA binding surface as it includes residues important for DNA recognition: Arg211, His212 and Ser215 from α-helix H3 (*Tsai et al., 2007*) and mutation of His212 also significantly hampers death induction by FOXO3 (*Figure 5a*). Furthermore, residues Arg222 and Trp234 from the β-strand S2 at the stem of wing W1 are involved in contact with the phosphate group of the DNA backbone. It is somehow surprising that compound S9 differentially binds to FOXO family members as demonstrated by FPAs (*Figure 1—figure supplement 2a*) and NMR (*Figure 2—figure supplement 2*) despite high homology. However, in a recent paper we demonstrated that the solution structures of DBD between these closely related FOXO proteins significantly differ thereby also explaining the differences in compound S9 – FOXO-DBD interaction (*Psenakova et al., 2019*).

Therefore, S9 binding could obstruct FOXO3-DBD interaction with the target DNA, thus blocking its transcriptional activity. The pharmacophore models used for virtual screening all reflected interactions with Arg211 and His212, in addition to (and dependent on the respective model) interactions with Asn208, Ser209, and Ser215. Model 1, which mapped compound S9, for example included hydrophobic interactions with Arg211 and His212 as well as hydrogen bonds with His212 and Asn208. The experimentally identified residues involved in binding overall align well with the in silico modeling and small re-arrangements of the proposed binding mode can be expected in solution.

Gene expression profiling provided an interesting three hours snap-shot on the mRNA changes induced by ectopic FOXO3 activation in presence or absence of compound S9. S9 strongly prevented the induction of FOXO3-induced genes or by its own repressed clusters of highly expressed FOXO3 targets, which is consistent with its interaction to a region in FOXO3-DBD including R211, H212 and S215 which all contribute to helix H3-DNA interaction (*Tsai et al., 2007*). On the other hand, S9 per se strongly induced a cluster of genes that was actively repressed by ectopically switched on FOXO3 suggesting that these genes are permanently repressed by endogenous FOXO3 in manner that requires the DNA-binding domain either for protein-DNA or protein-protein interaction and that this repression is abrogated by S9. A small cluster of genes was identified that was only moderately induced by ectopic FOXO3, but strongly up-regulated in presence of S9. One explanation for this phenomenon might be that DNA-binding independent transactivation by endogenous FOXO3 is further supported by S9 preventing intra-molecular protein-protein interaction of the C-terminal FOXO3 transactivation domain with the FOXO3-DBD as it involves the C-terminus of α-helix H3, which is part of the S9 binding pocket (*Wang et al., 2008*). These aspects concerning inhibition of intra- or inter-molecular protein-protein interaction at the FOXO3-DBD by S9 are investigated in currently ongoing studies.

Besides genome-wide regulation we tested the effect of S9 and S9OX on FOXO3 transcriptional targets. The BCL2-family members BIM and NOXA are mediators of FOXO3-induced intrinsic cell death and BIM critically triggers transitory accumulation of ROS at the mitochondria, which is

counteracted by the parallel induction of the detoxifying protein SESN3 (*Hagenbuchner et al., 2012*). DEPP1 sensitizes neuroblastoma cells for ROS and is strongly induced by FOXO3 via three functional FOXO consensus sequences in its promoter (*Salcher et al., 2014*; *Salcher et al., 2017*). We demonstrated that FOXO3 induction of these four *bona fide* targets was efficiently reduced by both S9 and S9OX on mRNA level as well as on protein expression level. To further prove direct inhibition of the FOXO3-DBD-DNA interaction in living cells, ChIP analyses on the four *bona fide* targets was performed which demonstrated that increased FOXO3 binding upon ectopic FOXO3-activation was almost completely abrogated by S9 (*Figure 4e*). This was consistent with results from primary FP analysis (*Figure 1f*) and dose-dependent repression of promoter activity by S9 and S9OX as demonstrated by luciferase reporter assay (*Figure 4f*). If H212 in the S9-interacting region of the FOXO3-DBD is mutated (*Figure 5a*) or endogenous FOXO3 is knocked out by CRISPR/Cas9 technology also the physiologic effects of S9 are abrogated (*Figure 5b–f*). This suggests that at least in the used model systems, S9 selectively acts via binding to the DBD of FOXO3. Consistent with the repression of target gene regulation, also FOXO3-induced ROS accumulation (*Hagenbuchner et al., 2012*; *Salcher et al., 2014*) was efficiently prevented by S9 and S9OX providing evidence that S9 directly interferes with physiological effects of FOXO3 in living cells. Of note S9 might also sterically hamper protein-protein interactions that take place at the FOXO3-DBD such as the binding of the tumor suppressor protein TP53 (*Rupp et al., 2017*; *Wang et al., 2008*). As affinity measurements using recombinant protein fragments provide only limited information about biologically effective concentrations, we generated magnetic 3D tumor spheres that allowed us on one hand to assess possible long term toxicity of S9/S9OX in 3D culture and on the other to test effective doses of these compounds. S9/S9OX neither reduced sphere-numbers nor diameter when spheres were treated 72 hr with 5 μM of these drugs, but efficiently prevented FOXO3-induced cell death. Interestingly, the growth-inhibitory effects of FOXO3 were only partially inhibited, which corresponds to the observation that gene-dosage effects of FOXO3 lead to, in part, contrary physiological outcomes (*Hagenbuchner et al., 2016b*).

FOXO transcription factors have been implicated in a plethora of different cellular functions, in cancer immunity where loss of FOXO3 exerts anti-cancer effects, but also hampers other arms of the immune system (*Luo and Li, 2018*), in neuronal development and plasticity (*McLaughlin and Broihier, 2018*), cancer angiogenesis and metabolism (*Yadav et al., 2018*). This excludes the permanent ablation of FOXO activity in mammals as a therapeutic option. More complexity is added by the fact that the DBD of FOXO transcription factors participates in intra-, but also intermolecular interactions with other cellular key regulators of death and longevity, such as TP53 (*Wang et al., 2008*) and thereby also affects drug resistance (*Rupp et al., 2017*; *Salcher et al., 2019*) or aging (*Baar et al., 2017*). Due to its mode of binding, S9 might well interfere with some of these protein-protein interactions. The effect of the small drug-like molecule S9 on specific tissues has to be investigated in detail to evaluate, whether S9 can serve as a chemical starting point for developing FOXO-regulatory compounds that steer distinct target gene subsets/functions of FOXO transcription factors. The advantage of small drug-like molecules such as S9 is the strict control of application-dose and -time and the fact that they are not immunogenic allowing repeated applications – so dose- and application time can be adjusted to damage cancer cells or boost anti-cancer immunity, but also limit unwanted side effects of FOXO-inhibition on stem cells and other somatic tissues.

## Materials and methods

Key Resources Table (see *Supplementary file 2*).

### Analysis of FOXO3-DNA interactions and generation of interaction maps

The experimentally observed interactions of the FOXO3 consensus sequence DNA strand and the FOXO3-DBD were extracted in LigandScout (*Wolber and Langer, 2005*) version 3.0 and represented by pharmacophore features. The pdb entry 2UZK (*Tsai et al., 2007*) was loaded into MOE version 11.2011. The protein was protonated using the default settings and interaction maps were generated using the N1:: (sp3 NH with lone pair), O (carbonyl oxygen), and DRY (hydrophobic) probes with −4 kcal/mol, −3.5 kcal/mol, and −2.5 kcal/mol cut-offs, respectively (*Figure 1c*).

## Docking, pharmacophore modeling, virtual screening, and selection of test compounds

GOLD (*Jones et al., 1997*) version 3.1 was used for docking. Hydrogen atoms were added to the protein and all water molecules and ligands were deleted. The area of 20.0 Å around His212 was defined as binding site and the GoldScore was used for scoring. Up to 10 different docking poses were reported for each input molecule and analyzed using LigandScout version 3.0. Pharmacophore models were created using LigandScout (*Wolber and Langer, 2005*) version 3.0 based on docking poses of molecules from Drugbank (*Wishart et al., 2008*) version 2.5 (exemplary docking poses of Drugbank molecules used for model generation, are shown in *Figure 1d*). Automatically generated models were manually refined to reflect interaction patterns observed in the crystal structure, highlighted by the interaction maps, or involved residues either crucial for binding or which are post-translational modification sites such as Asn208, Ser209, Arg211, His212, and Ser215. Compound libraries for virtual screening were generated by calculating up to 250 conformations for each molecule in the Specs (version April 2010, www.specs.net) and Maybridge (version 2010, www.maybridge.com) databases. Pharmacophore hits were docked into the binding site using the protocol described above and docking poses were analyzed using LigandScout version 3.0. 76 compounds, for which docking poses reflected the identified interaction patterns, were selected for experimental testing.

## Synthesis of S9 and S9OX

The synthesis of S9 and S9OX and all corresponding NMR spectra are described in *Supplementary file 1*.

## Expression and purification of recombinant FOXO3-DBD

Recombinant FOXO3-DBD protein was produced as previously described for FOXO4-DBD (*Boura et al., 2007*). Briefly, DNA encoding human FOXO3-DBD (residues 156–269) was ligated into the pGEX-6P-1 (GE Healthcare), using the BamHI and XhoI sites. FOXO3-DBD was expressed as N-terminal GST-tagged fusion protein in *E. coli* BL21(DE3). Protein expression in LB-media was induced by isopropyl β-D-1-thiogalactopyranoside for 18 hr at 20°C, and the protein was purified using Glutathione Sepharose 4 Fast Flow (Merck, Austria) according to a standard protocol. Protein was dialyzed against buffer containing 20 mM Tris-HCl, 100 mM NaCl, 1 mM EDTA, 1 mM DTT, 10% (wt/vol) glycerol at pH 7.5. The affinity tag was removed by PreScission Protease cleavage overnight at 4°C (10 U/mg recombinant protein). After the cleavage, FOXO3-DBD was purified by size-exclusion chromatography (HiLoad Superdex 75; GE Healthcare) in 20 mM phosphate buffer containing 50 mM KCl, 2 mM TCEP, and 10% (wt/vol) glycerol at pH 6.5. Isotopically labeled proteins for NMR experiments were expressed using the same procedure in minimal medium with [$^{15}$N]NH$_4$Cl and/or [$^{13}$C]glucose as the sole nitrogen and carbon sources, respectively, and purified as unlabeled FOXO3-DBD.

## Fluorescence polarization analyses (FPA)

To determine the binding of compound S9 to FOXO3-DBD in vitro FP-measurements were carried out in black 96-well plates with flat bottom (HVD Life Sciences, Vienna, Austria) in a chameleon plate reader (Hidex, Turku, Finland). For FP-assays, recombinant FOXO3 was only purified via affinity chromatography and gel-filtration chromatography. S9 or S9OX were added to 100 µl reaction mix containing 125 nM FOXO3-DBD and 25 nM FAM-labeled double strand FOXO3 consensus sequence oligonucleotides in assay buffer (20 mM TrisHCl, 100 mM NaCl, 1 mM EDTA, pH 7.5). Positive (only assay buffer and FAM-labeled oligonucleotide) and negative (FOXO3-DBD and FAM-labeled oligonucleotide) controls were analyzed on each plate. Millipolarization values (mP) were measured at an excitation wavelength of 485 nm and an emission wavelength of 530 nm.

## NMR data collection and analysis

NMR data were acquired on Bruker Avance III HD 600 MHz and 850 MHz spectrometers both equipped with a $^1$H/$^{13}$C/$^{15}$N cryoprobe at 25°C. The sequence-specific backbone resonance assignment was obtained using a series of standard triple-resonance spectra (HNCO, HN(CA)CO, HNCACB, and CBCA(CO)NH experiments). The obtained assignments were in a good agreement

with the sequence-specific backbone NMR assignment obtained for shorter FOXO3-DBD construct (*Wang et al., 2008*). Chemical shift perturbations (CSPs) were assessed using $^1$H-$^{15}$N heteronuclear single quantum coherence (HSQC) spectra. For these experiments, $^1$H-$^{15}$N HSQC spectra were collected for samples containing 250 µM FOXO3-DBD alone and in the presence of 1.5–7 mM S9OX. All NMR experiments were performed in buffer containing 20 mM sodium phosphate (pH 6.5), 50 mM KCl and 10% D$_2$O. STD-NMR experiments were acquired at 25°C using samples containing 15 µM unlabeled FOXO3-DBD in the presence of 1 mM ligand. 10% dimethyl sulfoxide (DMSO) was added to the buffer used in STD experiments with S9 to increase its solubility.

## Molecular docking

Docking experiments were performed using Autodock Vina (*Trott and Olson, 2010*). The solution structure of FOXO3-DBD (PDB ID: 2K86) and the S9 compound were modeled in Autodock Tools. The search space was defined as a 24 × 22 × 22 Å box centered on the helix H3 of the FOXO3-DBD. Flexible residues were selected based on CSP NMR analysis. The 20 lowest energy solutions were then analyzed and the lowest energy solution consistent with results from both STD and $^1$H-$^{15}$N HSQC experiments was selected as the final pose.

## Cell lines, culture conditions, and reagents

The neuroblastoma cell lines SH-EP/FOXO3 and NB15/FOXO3-GFP were cultured in RPMI1640 (Lonza, Basel, Switzerland) and Phoenix-AMPHO packaging cells and HEK293T cells in DMEM, respectively. All media contained 10% fetal bovine serum (GIBCO BRL, Paisley, UK), 100 U/ml penicillin, 100 µg/ml streptomycin and 2 mM L-glutamine (Sigma-Aldrich, Vienna, Austria) at 5% CO$_2$. All cultures were routinely tested for mycoplasma contamination using the Venor$^R$ GeM-mycoplasma detection kit (Minerva Biolabs, Germany). All reagents were purchased from (Merck, Austria) unless indicated otherwise.

## Transfections, Retroviral and lentiviral infection and CRISPR/Cas9 mediated knockout

For the transfection of retro- or lentiviral vectors into packaging cell lines the transfection reagent Lipofectamin 2000 (Thermofisher, USA) was used according to manufacturer's instructions. The retroviral vectors pBabe-puro-HA-FOXO3.A3.ER and pBabe-puro-HA-FOXO3.A3.ER.H212R were described before (*Czymai et al., 2010*). For CRISPR/Cas9 mediated knockout we used the lentiCRISPR/Cas9 v2 vector (Addgene #52961) (*Sanjana et al., 2014*) and inserted the gRNA sequences CACCGCCTGCCATATCAGTCAGCCG or CACCGCAGAGTGAGCCGTTTGTCCG to generate pLentiCRISPR-FOXO3-1 or pLentiCRISPR-FOXO3-2 vector, respectively. Retroviruses (for pBabe vectors) and lentiviruses (pLentiCRISPR vectors) were produced as previously described (*Hagenbuchner et al., 2012*). pBabe vectors for expression of conditional FOXO3 or FOXO3-H212R were infected into SH-EP cells generating the SH-EP/FOXO3 and SH-EP/FOXO3-H212R cell lines used in *Figure 5a*. pLentiCRISPR-FOXO3-1/2 were infected into SH-EP cells generating cell lines SH-EP/crispFOXO3#1 and SH-EP/crispFOXO3#2, respectively. Ectopic expression of FOXO3, FOXO3-H212R or deletion of endogenous FOXO3 was verified by immunoblot.

## Promoter activity

Reporter plasmids pGS-c10orf10/DEPP1-luc (*Salcher et al., 2014*), pGL2-luc-Bim (*Bouillet et al., 2001*) or pGL3-luc basic were transfected into target cells using JetPrime reagent (Polyplus, Berkeley, USA) and luciferase activity was measured using a Luciferase Assay System kit (Promega, Madison, USA) according to the manufacturer's instructions. The luminescence intensity was measured in a chameleon plate reader (Hidex, Turku, Finland).

## Spheroids

Spheroids were formed by magnetic bioprinting according to manufacturer's instructions (Pelobiotech GmbH, Germany). Spheroids were grown for 72 hr before treatment with 4OHT/4OHT + S9/S9OX. After 72 hr treatment compounds were removed by centrifugation and remaining spheroids were grown for one week. Finally, spheroids were collected and re-plated in 100 µl fresh media into white 96well plates for ATP measurement or clear plates for resazurin reduction. Size was monitored

regularly by live-cell microscopy. ATP-amount was measured by CellTiter-Glo-3D-cell-viability-assay according to manufacturer's instruction (Promega, Germany), resazurin salt reduction was assessed *via* fluorescence measurement in a chameleon plate reader (Hidex, Turku, Finland).

## Quantitative RT-PCR

To quantify mRNA-levels real-time qPCR was performed using Maxima-SYBR-Green-qPCR-Master-Mix (Thermofisher, Waltham, USA) and GAPDH as reference-gene as previously described (*Hagenbuchner et al., 2017*): total-RNA was isolated from $5 \times 10^6$ cells using TRI-Reagent (Merck, Vienna, Austria) and 1 µg was reverse-transcribed to cDNA using RevertAid-H-Minus-cDNA-Synthesis Kit (Thermo Scientific, Waltham, USA). Oligonucleotides for BIM, NOXA, DEPP1, SESN3, and GAPDH are listed in *Supplementary file 2*. qRT-PCR-reactions were performed in triplicates in a Bio-Rad-iCycler-instrument and repeated 3-times. After normalization to GAPDH, regulation was calculated between treated and untreated cells.

## Microarray expression profiling

Generation of the Affymetrix microarray data set was performed at the Expression Profiling Unit of the Medical University Innsbruck according to the manufacturer's protocols. Total RNA was prepared from SH-EP/FOXO3 cells treated for three hours with/without 50 nM 4OHT and/or 50 µM compound S9 and RNA quantity and quality was determined in a 2100 Bioanalyzer (Agilent Technologies, Palo Alto, USA). 250 ng of high quality RNA was processed, hybridized to twelve Human Genome U133 Plus 2.0 arrays (three independent biological replicates per treatment) and scanned according to manufacturer's instructions as described (*Schmidt et al., 2006*). The data analysis was performed in R (version). Raw data have been pre-processed using the GCRMA method (*Wu et al., 2004*).

## Immunoblot analyses

Total protein was prepared as described before (*Hagenbuchner et al., 2016a*). Proteins were separated by SDS-PAGE and blotted on nitrocellulose membrane (GE Healthcare, Chalfot, UK). After blocking, membranes were incubated with primary antibodies directed against Bim, Noxa, DEPP1, SESN3, FOXO3, GAPDH or αTubulin (*Supplementary file 2*), washed and incubated with horseradish-peroxidase conjugated secondary antibody (GE Healthcare, UK). The immunoblots were developed by enhanced chemiluminescence (Merck, Vienna, Austria) according to manufacturer's instructions and analyzed using an AutoChemi detection system (UVP, Cambridge, UK). Densitometry was performed using Labworks software version 4.5 (UVP, UK).

## Chromatin-immunoprecipitation (ChIP)

ChIP was performed using Magna-ChIP-Kit (Merck, Germany) as described (*Salcher et al., 2014*). 20 µl protein-G-magnetic-beads were coupled to 7.5 µl of FOXO3 antibody (Santa Cruz, Dallas, USA) or control-IgG and incubated with nuclear lysates of shredded DNA from $2 \times 10^7$ cells. After precipitation, protein was digested by proteinase-K and DNA was concentrated with ChIP-DNA-Clean- and -Concentrator-Kit (Zymo Research, USA). FOXO3-binding to DNA was quantified by qPCR using promoter-specific primers for FOXO3-targets.

## ROS measurements in living cells

Cells were grown on LabTek Chamber Slides (Nalge Nunc International, Rochester, USA), coated with 0.1 mg/ml collagen. ROS measurements were performed by incubating the cells with Mito-TrackerRed CM-$H_2$XROS (Thermofisher, USA) according to the manufacturer's instructions (final concentration 500 nM). Images were acquired with an Axiovert200M microscope (Zeiss, Vienna, Austria). Fluorescence intensity was quantified using Axiovision Software (Zeiss, Vienna, Austria) and relative ROS levels were expressed as % of untreated controls.

## Statistics and data analyses

Data were analyzed in GraphPad Prism seven software. Standard deviation (SD) and Student's t-test (two-tailed, with criteria of significance: *$p<0.05$; **$p<0.01$, ***$p<0.001$ and ****$p<0.0001$ or

$^{\#}p<0.05$; $^{\#\#}p<0.01$, and $^{\#\#\#}p<0.001$) or Mann Whitney U test ($^{***}p<0.001$) were calculated when applicable.

## Acknowledgements

We thank Anna Filipek and Patrick Markt for technical support, Vaclav Veverka and Lukas Vrzal for help with NMR measurements, Marc Schmidt and Feng Zhang for providing plasmids. This work was supported by COMET center of excellence ONCOTYROL,"Südtiroler Krebshilfe', 'Krebshilfe Südtirol Regenbogen', MFF-Tirol (Project 291), 'Tiroler Wissenschaftsförderung', Czech Science Foundation (Project 17–33854L), the 'Provita Kinderleukämiestiftung', the Austrian Science Fund (Project I3089-B28) and the 'Tirol-Kliniken GmbH'.

## Additional information

### Funding

| Funder | Grant reference number | Author |
| --- | --- | --- |
| Austrian Science Fund | I3089-B28 | Judith Hagenbuchner<br>Veronika Obsilova<br>Tomas Obsil<br>Michael J Ausserlechner |
| Grantová Agentura České Republiky | 17-33854L | Veronika Obsilova<br>Tomas Obsil |

The funders had no role in study design, data collection and interpretation, or the decision to submit the work for publication.

### Author contributions

Judith Hagenbuchner, Conceptualization, Formal analysis, Validation, Investigation, Visualization, Methodology, Project administration; Veronika Obsilova, Conceptualization, Funding acquisition, Investigation, Visualization, Methodology; Teresa Kaserer, Formal analysis, Investigation, Visualization, Methodology; Nora Kaiser, Formal analysis, Investigation, Methodology; Bettina Rass, Petra Obexer, Investigation; Katarina Psenakova, Investigation, Visualization; Vojtech Docekal, Miroslava Alblova, Klara Kohoutova, Investigation, Methodology; Daniela Schuster, Software, Supervision; Tatsiana Aneichyk, Formal analysis, Visualization; Jan Vesely, Supervision, Investigation, Methodology; Tomas Obsil, Conceptualization, Supervision, Funding acquisition, Investigation, Visualization, Project administration; Michael J Ausserlechner, Conceptualization, Formal analysis, Supervision, Funding acquisition, Validation, Investigation, Methodology, Project administration

### Author ORCIDs

Judith Hagenbuchner https://orcid.org/0000-0003-1396-3407
Veronika Obsilova https://orcid.org/0000-0003-4887-0323
Teresa Kaserer https://orcid.org/0000-0003-0372-1885
Katarina Psenakova http://orcid.org/0000-0001-8877-6599
Tomas Obsil https://orcid.org/0000-0003-4602-1272
Michael J Ausserlechner https://orcid.org/0000-0002-1015-2302

### Decision letter and Author response

Decision letter https://doi.org/10.7554/eLife.48876.sa1
Author response https://doi.org/10.7554/eLife.48876.sa2

## Additional files

### Supplementary files

- Supplementary file 1. Synthesis of S9 and S9OX.
- Supplementary file 2. Key Resources Table.

• Transparent reporting form

### Data availability

All data generated or analyzed during this study are included in manuscript and supporting files.

---

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
