## [Decision Letter]

**Acceptance summary:**

Both reviewers and myself judge the work to be important and timely for this field. Whether a DBD can be drugged is a burning question, and here you provide a very clear example of how drugs can be developed in this space. The work clearly warrants publication in *eLife* and as a result of the review process the added discussion and data on specificity is essential as drugging protein DNA interfaces is a major challenge in general for the community.

**Decision letter after peer review:**

Thank you for submitting your article "Modulating FOXO3 transcriptional activity by small, DBD-binding molecules" for consideration by *eLife*. Your article has been reviewed by two peer reviewers, and the evaluation has been overseen by a Reviewing Editor and Michael Eisen as the Senior Editor. The following individuals involved in review of your submission have agreed to reveal their identity: Mathias Francois (Reviewer #1); Ciara Metcalfe (Reviewer #2).

The reviewers have discussed the reviews with one another and the Reviewing Editor has drafted this decision to help you prepare a revised submission.

Summary:

Both reviewers and myself judge the work to be important and timely for this field. Whether a DBD can be drugged is a burning question, and here you provide a very clear example of how drugs can be developed in this space. The work clearly warrants publication in *eLife*, but it is also essential that you establish the specificity of the compound. Both reviewers and myself noted that the provided data do not unambiguously establish specificity. indeed, as noted by reviewer 2, "the NMR data (Figure 2—figure supplement 2) suggest an overlap in residues impacted by S9OX in FOXO1 and FOXO3, with distinct regions of the protein impacted in FOXO4" – suggesting that the DBD of all these family members may bind to S9OX, to some degree. These data do not support the statement that the NMR data "suggest selectivity of S9 and S9OX within the FOXO-family of transcription factors." It is therefore essential to explore the specificity in more detail. Exploring new chemical spaces to gain in specificity would likely be beyond the scope of this paper, though a great subject to discuss in this paper. However, a solid characterisation of the current 76 compound would establish a benchmark for future studies in this field.

Essential revisions:

I consider all the points raised by reviewers essential, with the exception of developing new variants of S9 that I consider super interesting but out of the scope of this paper. following is a summary of the concerns to address.

1) Selectivity issue: following is a list of the points raised by the two reviewers. I would like this point to be clarified in a revised work. Achieving selectivity is not important for publication but the selectivity must be known and documented.

2) Use of 4 OHT and doses of S9: several critical issues need to be documented here with identical doses of the compound and synergistic effects must be explored.

3) Cellular source of FOXO3: overexpression of FOXO3 is a problem and one would want to see some other approach. Reviewer 2 in her point 3 suggest a great possibility that would offer an orthogonal system to perform this study.

Reviewer #1:

This work by Psenakova and colleagues is really exciting and will directly contribute to the emerging and expanding field of drugging transcription. The molecular strategy chosen by the author is to target the DNA binding domain of FOXO3 transcription factor, using a pipeline that relies on: 1) an in silico-based approach (docking FOX/DNA to identify hot spot- generate 76 hits for in vitro testing via virtual screening using 2 chemical databases). 2) modelling of protein DNA interface to identify putative residues involved in small compound interaction. 3) combination of in vitro homogenous assay (FP) and in vitro cell-based assay with transcriptomics and tumour sphere assay as a readout for FOXO3 activity to validate inhibitor activity in living cells.

Overall the quality of the data is high and the manuscript clearly written. There is no doubt that the identified compound (S9) has some biological effects, at least in part mediated by FOXO3 inhibition, however the on-target engagement on FOXO3 lack some experimental evidences. With the current data set as it stands one cannot definitely come to the conclusion that S9 mostly acts via FOXO3 inhibition and interference of DNA binding.

Key experimental evidences missing to support the claims:

1) The rationale is to target the DNA binding activity of FOXO3. Aside from the FP data there is no evidence that S9 acts with this mode of action in cells to interfere with gene transcription. To address this the authors would need to perform ChiPSeq for FOXO3 in presence or absence of the compound and show that genome wide binding locations are interfered with by S9. At least the author should be able to provide some evidence on a subset of known FOXO3 direct target genes that FOXO3 binding is altered at known binding sites – This can be done by ChIP-qPCR analysis.

2) It would be helpful to perform thermal stability assay in presence of FOXO3 or a FOXO3 mutant (Arg 211, His 212, Ser215 amino acid thought to be involved with S9 interaction) with and without S9. This to further support the modelling data for compound/protein interaction.

3) It is difficult to understand how S9 achieve such a high level of selectivity to FOXO3 at least in FP while targeting such a highly conserved region as the DNA binding domain of FOX protein. Of note the FP experiments run for FOXO3 and other FOXO protein use different concentrations of S9 molecules (250nM for FOXOs vs. 500nM for FOXO3). To be able to compare selectivity it is necessary to perform the same dose response of S9 across all the FOXO protein.

4) The work rely solely on the use of 1 compound, it is therefore difficult to assess the level of selectivity of this molecular space. How are other compounds closely related (with a similar pharmacophore) but with no predicted DNA binding activity behave (at least in FP). Can S9 affect FOXM1 DNA binding activity? These data would help to assess specificity and efficacy of S9 as a protein/DNA disruptor.

5) It Is possible that S9 and 4-OHT have a synergic or additive effect on FOXO3 gene regulatory network. The analysis of the RNAseq data set should revised and include an analysis of DEseq for -4OHT with -4OHT+S9. This would be useful to assess the effect of the inhibitor on endogenous FOXO3 activity.

6) The authors highlight that the SH-EP neuroblastoma cell line used in this work has some level of FOXO3 activity. In the tumour sphere assay there is no effect of S9 in condition without 4OHT. It seems that S9 only works in over-expression condition- if there is endogenous FOXO3 expression in SHEP cells why is there no effect of S9?

Reviewer #2:

Hagenbuchner et al. describe inhibition of the transcription factor FOXO3 by small molecule targeting of its DNA-binding domain. Therapeutic targeting of transcription factors is highly desirable, but remains a significant challenge. Practical advancements in this area are thus potentially impactful, and the topic of this study is thus relevant for publication in *eLife*. However, there are some key elements of the work where improvements are warranted:

1) – Given that a major concern of small molecule perturbation of DNA-binding is the specificity of this approach, it is important to clearly establish selectivity of S9/S9OX, across the FOXO family and beyond. While the fluorescence polarization assay measuring binding of FOXO-DBD to labelled oligo provides some evidence of potential selectivity (though the result for FOXO4 is borderline), the NMR data (Figure 2—figure supplement 2) suggests an overlap in residues impacted by S9OX in FOXO1 and FOXO3, with distinct regions of the protein impacted in FOXO4 – suggesting that the DBD of all these family members may bind to S9OX, to some degree. These data do not support the statement that the NMR data "suggests selectivity of S9 and S9OX within the FOXO-family of transcription factors."

2) – Given the above, it would be useful to see the primary biochemical validation data (i.e. fluorescence polarization assay) for the 76 virtual hits that were experimentally tested, to understand the dynamic range, sensitivity and specificity of the in vitro screening assay e.g. what did an unconfirmed hit look like, vs. S9? Ideally, a secondary screen would have been performed against an unrelated DBD – was this done?

3) – The mRNA expression data showing attenuation of 4OHT-induced transcription by S9(OX), and the CHIP data showing a prevention of 4OHT-induced DNA binding of FOXO3 are highly encouraging. However, a limitation is that much of the functional data is in the context of exogenously expressed, ER-tagged, mutant FOXO3. Activity states of endogenous FOXO3 can be "toggled" using PI3K/AKT inhibitors (e.g. Santo et al., Cancer Research, 2013). An assessment of how S9(OX) impacts endogenous FOXO3-mediated transcription and DNA-binding under these conditions would provide more compelling support for S9 mechanism.

4) – For the viability experiments on the NB15 cells, the S9 dose is dropped to 5µM, while the FOXO3-specific pathway assessments were conducted at 50µM – it would be worthwhile to show a dose response of both the transcriptional and physiological effects to demonstrate that the physiological consequences of S9-treatment are due to an on-target effect on transcription i.e. the transcriptional and physiological phenotypes should occur/arise at similar drug concentrations.

---

## [Author Response]

Summary:Both reviewers and myself judge the work to be important and timely for this field. Whether a DBD can be drugged is a burning question, and here you provide a very clear example of how drugs can be developed in this space. The work clearly warrants publication in eLife, but it is also essential that you establish the specificity of the compound. Both reviewers and myself noted that the provided data do not unambiguously establish specificity. indeed, as noted by reviewer 2, "the NMR data (Figure 2—figure supplement 2) suggest an overlap in residues impacted by S9OX in FOXO1 and FOXO3, with distinct regions of the protein impacted in FOXO4" – suggesting that the DBD of all these family members may bind to S9OX, to some degree. These data do not support the statement that the NMR data "suggest selectivity of S9 and S9OX within the FOXO-family of transcription factors." It is therefore essential to explore the specificity in more detail. Exploring new chemical spaces to gain in specificity would likely be beyond the scope of this paper, though a great subject to discuss in this paper. However, a solid characterisation of the current 76 compound would establish a benchmark for future studies in this field.

We now revised Figure 1 showing the whole *in silico* screening strategy that was previously split between Figure 1 and Figure 1—figure supplement 1 and now added the screening data (fluorescence polarization assay (FPA) and flow cytometric analyses) for the 76 candidate compounds as novel Figure 1—figure supplement 1. We also explain there, how the different compounds were clustered according to the combined FP- and cytometry live/dead screening results. We replaced also the previous FP graph (now Figure 1F) and cell based validation assay (now Figure 1G) by experiments showing dose dependent effects of S9 and S9ox. In Figure 1G we used a constant concentration of 4OHT to activate ectopic FOXO3, varied the concentration of S9 from 3.1 µM to 50 µM and nicely demonstrate that S9 and S9OX inhibit cell death in a dose dependent manner.

Essential revisions:I consider all the points raised by reviewers essential, with the exception of developing new variants of S9 that I consider super interesting but out of the scope of this paper. following is a summary of the concerns to address.1) Selectivity issue: following is a list of the points raised by the two reviewers. I would like this point to be clarified in a revised work. Achieving selectivity is not important for publication but the selectivity must be known and documented.2) Use of 4 OHT and doses of S9: several critical issues need to be documented here with identical doses of the compound and synergistic effects must be explored.

Now, we also performed a dose-response analysis with constant concentrations of S9 and varying doses of 4OHT, as suggested by the reviewers. This demonstrates that the compounds S9 and S9OX do not induce programmed cell death per se and that above a certain level of activation of ectopic FOXO3 (i.e. approx. 20 nM 4OHT) cell death induction reaches a plateau in this model system within 48 hours. As S9 inhibits FOXO3 and also ectopic FOXO3-induced cell death triggered by 4OHT, synergistic effects between 4OHT and S9 cannot be detected.

3) Cellular source of FOXO3: overexpression of FOXO3 is a problem and one would want to see some other approach. Reviewer 2 in her point 3 suggest a great possibility that would offer an orthogonal system to perform this study.

Selectivity issue and cellular source of FOXO3:

We agree with the editors and the reviewers that the selectivity issue / possible off target effects of S9 were not fully investigated in the previous manuscript. In addition, the reviewers noted correctly that all biological studies were performed using a conditional ectopically expressed FOXO3 construct that might behave different to endogenous FOXO3.

We tried to address these questions in detail and generated cell lines where endogenous FOXO3 was homozygous deleted by CRISPR/Cas 9 technology and cell lines carrying a conditional FOXO3 allele with a mutation at histidine H212 (H212 is central to the S9 binding pocket and showed a significant shift upon S9 binding, see Figure 2B and C). Cell biological experiments demonstrate that FOXO3 is essential for the induction of target genes Bim, DEPP1 and SESN3 under low serum conditions or genotoxic stress (Figure 5B-F). Upon deletion of endogenous FOXO3 the compound S9 does not repress above FOXO3 targets in response to serum withdrawal and / or etoposide treatment any longer. This proves that S9 exerts its inhibitory effect exclusively via endogenous FOXO3. Importantly, mutation of H212 markedly inhibits the death-inducing abilities of ectopic FOXO3, but completely abrogates S9 effects suggesting that H212 is essential for S9 – FOXO3 interaction. We now added a complete new Figure 5 to the manuscript showing these data.

To address the relevance of endogenous FOXO3 for the growth of tumor spheroids we cultured NB15/crispFOXO3 cells (FOXO3 knockout) for three days with/without S9 and observed that knockout of endogenous FOXO3 has no significant effect on viability of 3D tumor spheroids.

To address possible binding of S9 beyond FOXO family members we performed FPAs with recombinant FOXM1 protein (Figure 1—figure supplement 2B) and the suggested Differential Scanning Fluorimetry (DSF) assays (reviewer #1, point 3) with the newly generated FOXO3 mutant (Arg211, His212 and Ser215). Like for the FOXO-family member FOXO6, S9 did not affect FOXM1-DBD – DNA interaction as measured by FPA, suggesting that S9 is not effective on FOXM1. DSF analyses did not provide clear results which might be explained by too low sensitivity of this method. Nevertheless, a subtle stabilization was observed for the wildtype and a subtle destabilisation for the triple mutant.

In an ongoing project we are also investigating the chemical spaces and are trying to generate improved chemical derivatives.

Reviewer #1:
*This work by Psenakova and colleagues is really exciting and will directly contribute to the emerging and expanding field of drugging transcription. The molecular strategy chosen by the author is to target the DNA binding domain of FOXO3 transcription factor, using a pipeline that relies on: 1) an in silico-based approach (docking FOX/DNA to identify hot spot- generate 76 hits for* in vitro *testing via virtual screening using 2 chemical databases). 2) modelling of protein DNA interface to identify putative residues involved in small compound interaction. 3) combination of* in vitro*homogenous assay (FP) and* in vitro cell-based assay with transcriptomics and tumour sphere assay as a readout for FOXO3 activity to validate inhibitor activity in living cells.Overall the quality of the data is high and the manuscript clearly written. There is no doubt that the identified compound (S9) has some biological effects, at least in part mediated by FOXO3 inhibition, however the on-target engagement on FOXO3 lack some experimental evidences. With the current data set as it stands one cannot definitely come to the conclusion that S9 mostly acts via FOXO3 inhibition and interference of DNA binding.Key experimental evidences missing to support the claims:1) The rationale is to target the DNA binding activity of FOXO3. Aside from the FP data there is no evidence that S9 acts with this mode of action in cells to interfere with gene transcription. To address this the authors would need to perform ChiPSeq for FOXO3 in presence or absence of the compound and show that genome wide binding locations are interfered with by S9. At least the author should be able to provide some evidence on a subset of known FOXO3 direct target genes that FOXO3 binding is altered at known binding sites – This can be done by ChIP-qPCR analysis.

Figure 4E shows a ChIP-qPCR analysis for the *bonafide* FOXO3 target genes Bim, Noxa, SESN3 and DEPP1 demonstrating that S9 directly interferes with DNA binding at the genomic level.

2) It would be helpful to perform thermal stability assay in presence of FOXO3 or a FOXO3 mutant (Arg 211, His 212, Ser215 amino acid thought to be involved with S9 interaction) with and without S9. This to further support the modelling data for compound/protein interaction.

As suggested by the reviewer, we prepared the FOXO3-DBD triple mutant (R211A, H212A, S215A) and compared its thermal stability with FOXO3-DBD WT in the absence and presence of 500 μm S9OX using differential scanning fluorimetry (DSF, thermal shift assay). As noticed, the sensitivity of this technique is not high enough to detect weak interaction between FOXO3-DBD and S9. Nevertheless, we observed subtle stabilization in the case of WT and subtle destabilization for the mutant, however in both cases observed changes in Tm were not statistically significant. Listed values are mean +/- SD from six measurements. Protein concentration was 15 µM. In addition, we also generated cell lines expressing a conditional FOXO3-H212R mutant allel and tested the effect of this mutant in presence or absence of S9. As shown in Figure 5A, this mutant has residual cell death inducing activity, which is not affected by S9 or S9OX any longer. As death induction in this cell model depends on the transcriptional induction of Bim (Hagenbuchner et al., 2012) these data suggest that the residual transcriptional activity of this mutant is not affected by S9 due to significant changes in the S9 binding pocket that hamper S9 binding.

3) It is difficult to understand how S9 achieve such a high level of selectivity to FOXO3 at least in FP while targeting such a highly conserved region as the DNA binding domain of FOX protein.

Although the DBD is highly conserved within the FOXO family members detailed analyses of NMR solution structures of FOXO1-DBD, FOXO3-DBD and FOXO4-DBD demonstrates that the positions of helices H1, H2 and H3 and the interactions between hydrophobic residues at the interface between the N-terminal segment, the H2-H3 loop and the recognition helix H3 differ within the FOXO protein family members (Psenakova et al., 2019). This may explain why compound S9 shows different effects in comparative FPAs and targets FOXO family members to different extent.

Of note the FP experiments run for FOXO3 and other FOXO protein use different concentrations of S9 molecules (250nM for FOXOs vs. 500nM for FOXO3). To be able to compare selectivity it is necessary to perform the same dose response of S9 across all the FOXO protein.

We would like to point out that in Figure 1—figure supplement 2A the identical conditions (i.e. concentrations of FOXO-proteins, FAM-labelled oligonucleotide and S9 (250nM)) were used for all proteins. Only in the FPA shown in previous Figure 1 we had a concentration of 500 nM. Nevertheless, we now also replaced the FPA in Figure 1 by a graph showing dose-dependent effects of S9 (range between 250 nM and 1 µM).

4) The work rely solely on the use of 1 compound, it is therefore difficult to assess the level of selectivity of this molecular space. How are other compounds closely related (with a similar pharmacophore) but with no predicted DNA binding activity behave (at least in FP).

We now added, as also requested by reviewer #2, the data on the FPA / live/dead validation process as Figure 1—figure supplement 1 to the manuscript showing 76 candidate compounds identified by pharmacophore modelling / in silico screening. Despite all of these candidates were selected by the pharmacophore models only 11 of them displaced the consensus oligonucleotide in the FPA. In an ongoing project we are now investigating in detail the chemical spaces that define binding of S9 and we also try to improve the affinity / biological efficacy.

Can S9 affect FOXM1 DNA binding activity? These data would help to assess specificity and efficacy of S9 as a protein/DNA disruptor.

We expressed and purified recombinant FOXM1-DBD (222-360) protein and tested S9 at a concentration of 500 µM in parallel in FOXO3-FPA and FOXM1-FPA. As shown in Figure 1—figure supplement 2B, similar to FOXO6 the compound S9 does not interfere with binding of FOXM1 to its consensus sequence.

5) It Is possible that S9 and 4-OHT have a synergic or additive effect on FOXO3 gene regulatory network. The analysis of the RNAseq data set should revised and include an analysis of DEseq for -4OHT with -4OHT+S9. This would be useful to assess the effect of the inhibitor on endogenous FOXO3 activity.

We addressed this issue in cell biological experiments by either constant 4OHT concentrations and varying S9 or S9OX (Figure 1G) or constant S9 concentration and varying 4OHT. Of note, as S9 or S9OX do not affect viability per se, on a physiologic level no synergism between 4OHT and S9 can be observed. We now mentioned in the text that S9 by its own caused repression of 412 genes (subsection “Compounds S9 and S9OX affect the FOXO3 transcriptional program”. To further assess the effect of S9 on endogenous FOXO3 we generated FOXO3 knock out cells using CRISPR/Cas9 technology and analyzed the effect of serum withdrawal and/or etoposide treatment on target gene regulation in presence or absence of S9 and S9OX. The combined results are shown in Figure 5B-F and demonstrate that knockout of endogenous FOXO3 also abrogates the effects of S9 on gene regulation of these FOXO target proteins in response to growth factor withdrawal and/or genotoxic stress.

6) The authors highlight that the SH-EP neuroblastoma cell line used in this work has some level of FOXO3 activity. In the tumour sphere assay there is no effect of S9 in condition without 4OHT. It seems that S9 only works in over-expression condition- if there is endogenous FOXO3 expression in SHEP cells why is there no effect of S9?

The heatmap in Figure 3A demonstrates that S9 affects the gene-regulation patterns also in the absence of 4OHT (comparison between DMSO and S9) but apparently this does not affect genes that are critical for survival and/ or proliferation. The fact that FOXO3 is dispensable at normoxic conditions (Hagenbuchner et al., 2017) for the survival of SH-EP as well as of NB15 neuroblastoma cells that we used for formation of tumor spheres is further demonstrated by the fact that FOXO3 knockout cells show no reducedproliferation or viability (Figure 5B-F). In Figure 5 we also demonstrate that compound S9 and S9OX exert pronounced effects on endogenous gene regulation under conditions of serum withdrawal and/or genotoxic stress – so the compound acts also on endogenous FOXO3.

Reviewer #2:Hagenbuchner et al. describe inhibition of the transcription factor FOXO3 by small molecule targeting of its DNA-binding domain. Therapeutic targeting of transcription factors is highly desirable, but remains a significant challenge. Practical advancements in this area are thus potentially impactful, and the topic of this study is thus relevant for publication in eLife. However, there are some key elements of the work where improvements are warranted:1) Given that a major concern of small molecule perturbation of DNA-binding is the specificity of this approach, it is important to clearly establish selectivity of S9/S9OX, across the FOXO family and beyond. While the fluorescence polarization assay measuring binding of FOXO-DBD to labelled oligo provides some evidence of potential selectivity (though the result for FOXO4 is borderline), the NMR data (Figure 2—figure supplement 2) suggests an overlap in residues impacted by S9OX in FOXO1 and FOXO3, with distinct regions of the protein impacted in FOXO4 – suggesting that the DBD of all these family members may bind to S9OX, to some degree. These data do not support the statement that the NMR data "suggests selectivity of S9 and S9OX within the FOXO-family of transcription factors."

*2) Given the above, it would be useful to see the primary biochemical validation data (i.e. fluorescence polarization assay) for the 76 virtual hits that were experimentally tested, to understand the dynamic range, sensitivity and specificity of the* in vitro *screening assay e.g. what did an unconfirmed hit look like, vs. S9? Ideally, a secondary screen would have been performed against an unrelated DBD – was this done?*

Following the reviewer’s suggestions we now included the results of the primary biochemical validation screen and the live/dead screen of the 76 candidate compounds as Figure 1—figure supplement 1. Although the FPA demonstrated that DNA-interaction of the FOXO family member FOXO6 was not affected by S9 we also performed FPA assays with recombinant FOXM1-DBD protein. The data suggest that S9 does not affect the interaction of FOXM1-DBD with its cognate consensus sequence (Figure 1—figure supplement 2B). In addition, we also generated cell lines with a conditional FOXO3-H212R allele. In this FOXO3 allele the H212, which had the most pronounced shift in NMR upon S9 binding, is mutated. This mutant shows strongly reduced, but still visible induction of cell death (Figure 5A), which depends on the induction of Bim in these neuroblastoma cells (Hagenbuchner et al., 2012). Importantly, in these cells the addition of S9 does not affect cell death at all, suggesting that the loss of H212 in FOXO3 prevents S9 – FOXO3-DBD interaction.

To address the selectivity issue we knocked out endogenous FOXO3 by CRISPR/Cas9 technology, applied growth factor withdrawal and/or genotoxic stress by etoposide treatment which both trigger activation of endogenous FOXOs (Hagenbuchner et al., 2012; Hagenbuchner et al., 2013). S9 and S9OX block the activation of reporter plasmids for Bim and DEPP1 promoter (Figure 5B, C), as well as the induction of endogenous Bim, DEPP1 and SESN3 (Figure 5D-F). Importantly, in FOXO3 knockout cells S9 did not show any inhibitory effect, thereby demonstrating that S9 / S9OX acts onto stress- and growth factor withdrawal-induced genes via FOXO3.

3) The mRNA expression data showing attenuation of 4OHT-induced transcription by S9(OX), and the CHIP data showing a prevention of 4OHT-induced DNA binding of FOXO3 are highly encouraging. However, a limitation is that much of the functional data is in the context of exogenously expressed, ER-tagged, mutant FOXO3. Activity states of endogenous FOXO3 can be "toggled" using PI3K/AKT inhibitors (e.g. Santo et al., Cancer Research, 2013). An assessment of how S9(OX) impacts endogenous FOXO3-mediated transcription and DNA-binding under these conditions would provide more compelling support for S9 mechanism.4) For the viability experiments on the NB15 cells, the S9 dose is dropped to 5µM, while the FOXO3-specific pathway assessments were conducted at 50µM – it would be worthwhile to show a dose response of both the transcriptional and physiological effects to demonstrate that the physiological consequences of S9-treatment are due to an on-target effect on transcription i.e. the transcriptional and physiological phenotypes should occur/arise at similar drug concentrations.

This is difficult to address because the physiological outcome of FOXO3 activation is not monogenic but a balance in the regulation of different genes that are also activated to different extent (Hagenbuchner and Ausserlechner, 2013). Therefore, the induction (or repression by S9 / S9OX) of one gene might not correlate directly with the physiologic outcome. To demonstrate dose-dependent inhibition of FOXO3-induced cell death we now included a dose-response curve as Figure 1G. We hope that the reviewer agrees with us that the selective knockout of FOXO3 is a good strategy to demonstrate that S9 acts via FOXO3-induced transcription. Please note that serum withdrawal also increases the expression of FOXO3 target genes, primarily involved in stress-detoxification (Figure 5D-F) in the FOXO3 knockout background suggesting that another serum responsive regulator induces the transcription and that this induction is not affected by S9 or S9OX.

References:

Hagenbuchner, J., and M. J. Ausserlechner. (2013). Mitochondria and FOXO3: breath or die. Frontiers in Physiology 4:147.